# Ultrathin positively charged electrode skin for durable anion-intercalation battery chemistries

Davood Sabaghi[1,11], Zhiyong Wang[1,2,11], Preeti Bhauriyal [3,11], Qiongqiong Lu [4], Ahiud Morag[1], Daria Mikhailovia[4], Payam Hashemi [1,2], Dongqi Li[1], Christof Neumann [5], Zhongquan Liao[6], Anna Maria Dominic[1], Ali Shaygan Nia[1,2], Renhao Dong [1,7] ✉, Ehrenfried Zschech[8], Andrey Turchanin [5], Thomas Heine [3,9,10], Minghao Yu [1] ✉ & Xinliang Feng [1,2] ✉

The anion-intercalation chemistries of graphite have the potential to construct batteries with promising energy and power breakthroughs. Here, we report the use of an ultrathin, positively charged two-dimensional poly(pyridinium salt) membrane (C2DP) as the graphite electrode skin to overcome the critical durability problem. Large-area C2DP enables the conformal coating on the graphite electrode, remarkably alleviating the electrolyte. Meanwhile, the dense face-on oriented single crystals with ultrathin thickness and cationic backbones allow C2DP with high anion-transport capability and selectivity. Such desirable anion-transport properties of C2DP prevent the cation/solvent co-intercalation into the graphite electrode and suppress the consequent structure collapse. An impressive $PF_6^-$-intercalation durability is demonstrated for the C2DP-covered graphite electrode, with capacity retention of 92.8% after 1000 cycles at 1 C and Coulombic efficiencies of > 99%. The feasibility of constructing artificial ion-regulating electrode skins with precisely customized two-dimensional polymers offers viable means to promote problematic battery chemistries.

Low-cost, environment-friendly, and redox-amphoteric graphite has been recognized as a versatile battery electrode material in light of its ability to host both cationic and anionic intercalants[1–4]. Notably, the anion-intercalation reactions of graphite are featured by the high redox potential (on average >1.5 V vs. standard hydrogen electrode) and ultrafast kinetics, thus raising a variety of sustainable battery

concepts in the latest years[5–7]. For example, by replacing conventional Li⁺-hosting cathodes in Li-ion batteries with the anion-hosting graphite cathode (e.g., hexafluorophosphate ($PF_6^-$)[2], bis(trifluoromethanesulfonyl) imide (TFSI⁻)[8], and bis(fluorosulfonyl)imide (FSI⁻)[9]), the so-called dual-ion batteries (DIBs) have been demonstrated as robust alternatives for high-voltage (>4.5 V), fast-rate, and

[1]Center for Advancing Electronics Dresden (cfaed) & Faculty of Chemistry and Food Chemistry, Technische Universität Dresden, Mommsenstraße 4, 01062 Dresden, Germany. [2]Max Planck Institute of Microstructure Physics, D-06120 Halle (Saale), Germany. [3]Theoretical Chemistry, Technische Universität Dresden, 01062 Dresden, Germany. [4]Leibniz Institute for Solid State and Materials Research (IFW), 01069 Dresden, Germany. [5]Institute of Physical Chemistry, Friedrich Schiller University Jena, 07743 Jena, Germany. [6]Fraunhofer Institute for Ceramic Technologies and Systems (IKTS), Dresden 01109, Germany. [7]Key Laboratory of Colloid and Interface Chemistry of the Ministry of Education, School of Chemistry and Chemical Engineering, Shandong University, Jinan 250100, China. [8]Faculty of Chemistry, University of Warsaw, ul. Żwirki i Wigury 101, Warsaw 02-089, Poland. [9]Institute of Resource Ecology, Helmholtz-Zentrum Dresden-Rossendorf, Leipzig Research Branch, 04316 Leipzig, Germany. [10]Department of Chemistry, Yonsei University, Seodaemun-gu Seoul 120-749, Korea. [11]These authors contributed equally: Davood Sabaghi, Zhiyong Wang, Preeti Bhauriyal. ✉e-mail: renhaodong@sdu.edu.cn; minghao.yu@tu-dresden.de; xinliang.feng@tu-dresden.de

large-scale energy storage applications[10]. The DIB concept, together with the anion-intercalation graphite chemistries, has also been extended to other cation systems (e.g., alkaline Na+, K+ and multivalent Zn2+, Ca2+, and Al3+), elaborately bypassing the sluggish cathode chemistries associated with those large-size or multivalent cations beyond lithium[11–13]. Nevertheless, DIBs suffer from a mismatching problem between the operating cutoff voltage of batteries (e.g., >5 V for Li-based DIBs) and the stable potential window of electrolytes (e.g., <5 V vs. Li+/Li for conventional carbonate- and ether-based electrolytes)[14,15]. The high operating voltage of DIBs leads to low Coulombic efficiency (<90% at early cycling) and eventually battery failure (after a few hundred cycles at a low current density), severely restricting the practical deployment of DIBs. There are two main reasons for the unsatisfactory battery performance (Fig. 1a). The electrolyte molecules tend to be oxidized at the exposed active graphite edge under the high voltage, causing the serious electrolyte decomposition and the formation of thick and resistant cathode-electrolyte interphase[16–18]. Meanwhile, cation/solvent co-intercalation, together with the gas release due to the electrolyte decomposition, leads to the fast structural degradation of the graphite cathode[18–20].

For Li-ion and Li-metal batteries, it has been widely demonstrated that the presence of a stable solid electrolyte interface (SEI) holds a vital role in promoting the reversibility and durability of anode chemistries[21–24]. Effective SEI is known as a uniform and dense passivation layer on an anode surface, which is composed of intricate compounds arising from the reduction reactions of the electrolyte (including the salt, solvent, and additive)[23,25–27]. Such a passivation layer is electrically insulating to prevent continuous electrolyte reduction. Meanwhile, it is ionic conductive to allow the charge carrier ions to pass through the SEI layer and into the electrode. Following a similar principle, careful design of a solid graphite-electrolyte interface should, in principle, address the inherent durability problem of the anion-intercalation graphite chemistries, which, however, has been seldomly investigated. An obvious obstacle lies in the difficulty of forming uniform and stable cathode interfaces through the oxidation reactions of electrolytes, as is the case with the conventional Li-ion battery cathodes[23]. Although artificial cathode interfaces can be possibly obtained through adding electrolyte additives or preforming on the electrode prior to the battery assembly, the multicomponent and dense interfaces typically formed could lead to the problematic transport of anions consisting of multiple atoms and complex configurations, let alone the desirable anion-selective transport[28,29].

Here, we demonstrate the use of ultrathin positively charged two-dimensional poly(pyridinium salt) membrane (denoted C2DP) as the artificial skin for the graphite cathode, which tackles the inherent reversibility and durability issues of the anion-intercalation chemistries (Fig. 1b). Specifically, large-area C2DP (28 cm2), fabricated by a facile and upscalable on-water surface synthesis approach, can be conformally coated on the graphite electrode. Like SEI, electron-insulating C2DP provides a passivation layer at the electrode-electrolyte interface, avoiding electrolyte decomposition and the consequent formation of the anion-blocking layer. Meanwhile, the densely equipped single-crystal domains with an ultrathin thickness

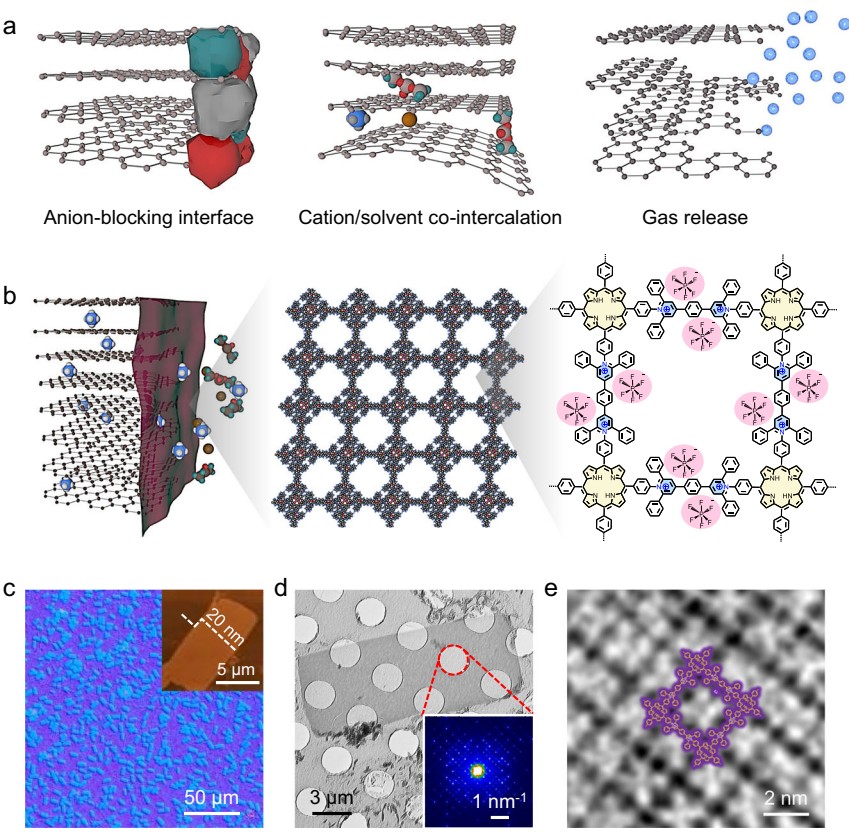

**Fig. 1 | Anion-intercalation chemistries of graphite and morphological characterizations of C2DP. a** Schematic illustration showing challenges associated with the anion-intercalation chemistries of graphite. **b** Functions of C2DP as the electrode skin in promoting the anion-intercalation chemistries. **c** Optical microscopy image of C2DP. The inset shows an atomic force microscopy image of a single crystal (size of 60 μm2), and the height profile along the white line indicate that the thickness of the single crystal is 20 nm. **d** TEM image of C2DP. The inset shows the selected area electron diffraction pattern of a single crystal (the red-cycle zone). **e** High-resolution TEM image of C2DP with the structural model overlaid.

(20 nm), a well-defined cationic backbone, and quasi-1D nanochannels confer C2DP with a high anion-transport capability (PF$_6^-$ diffusivity: ~10$^{-7}$ cm$^2$ s$^{-1}$) and selectivity (PF$_6^-$ transference number increased from 0.17 for the polypropylene separator to 0.51 for the C2DP-loaded polypropylene separator). When used as the graphite electrode skin, C2DP effectively screens cation/solvent from intercalating into graphite and thus suppresses the structure collapse of graphite grains. Besides, the 'soft' polymerized structure with irreversible pyridinium linkages empowers C2DP with good chemical stability and mechanical robustness to adapt to the complex battery environment and withstand the electrode volume change. Consequently, the C2DP-covered graphite cathode (denoted C2DP-G) achieves an impressive PF$_6^-$-intercalation durability with the capacity retention of 92.8% after 1000 cycles at 1 C and Coulombic efficiencies of >99%, which contrasts with the pronounced performance degradation of the pristine graphite cathode after 300 cycles. The long-term durability of the C2DP-G electrode represents the best among the previously reported anion-intercalation chemistries of graphite in organic electrolytes. Furthermore, the protective function of C2DP also works for the TFSI$^-$-intercalation and FSI$^-$-intercalation chemistries of graphite, indicating the universal effect of C2DP for various anion-related battery chemistries.

## Results

### Synthesis and characterization of C2DP

Two-dimensional (2D) polymer membranes show controllable and exotic physicochemical properties, in light of their ultrathin thickness (typically <50 nm) and flexible structural tunability by variation of covalently bonded building blocks. The feasible regulation of ion behavior (e.g., ion desolvation, anion-cation dissociation, and ion-selective transport) offers precisely customized 2D polymer membranes with intriguing but unrealized promise as artificial interfaces for diverse battery electrodes. To allow the use as the effective interface for anion-intercalation graphite, the ideal 2D polymer membrane is supposed to have large-scale continuity (at least cm$^2$ scale for device integration), regular pore channels, and abundant anion-transport sites on the polymer backbone. Based on this design principle, a C2DP membrane (up to ~28 cm$^2$, Supplementary Fig. 1) was synthesized via the surfactant monolayer-assisted interfacial synthesis approach (Supplementary Fig. 2), using the Katritzky reaction of 5,10,15,20-(tetra-4-aminophenyl) porphyrin (compound 1) and 1,4-phenylene-4,4'-bis (2,6-diphenyl-4-pyrylium) tetrafluoroborate (compound 2)[27,30]. The high reaction selectivity of Katritzky reaction in our interfacial synthesis approach was verified by the model Katritzky reaction with 5-(4-aminophenyl) −10,15,20-(triphenyl)porphyrin and 2,4,6-triphenylpyrylium tetrafluoroborate under the similar reaction condition. The performed matrix-assisted laser desorption/ionization−time-of-flight mass spectrometry (MALDI-TOF MS) result of the formed product on the water surface indicates the presence of the pyridinium target without detectable by-products (Supplementary Fig. 3). Moreover, the synthesis procedure of C2DP was extensively optimized to reach a high yield rate of >95 %. The C2DP membrane can be easily transferred onto targeted substrates for further characterization or investigation (e.g., Si wafer, Supplementary Fig. 4). A plentiful of single crystals with a domain size of 40-100 μm$^2$ and a thickness of ~20 nm are observed in the optical microscopy image of C2DP (Fig. 1c). Moreover, the highly ordered square lattice with a face-on orientation at the crystalline domain was straightforwardly visualized by the transmission electron microscopy (TEM) images (Fig. 1d, e) and the grazing incidence wide-angle X-ray scattering pattern (Supplementary Fig. 5). In addition, the efficient conversion of the amino groups in compound 1 to the positively charged pyridinium rings of C2DP

was confirmed by Fourier-transform infrared (Supplementary Fig. 6) and Raman (Supplementary Fig. 7) spectra.

### None-degraded capacity and rate performance

Prior to evaluating the role of C2DP as the artificial skin for the graphite electrode, we first examined the stable potential window of C2DP. As reflected by the linear sweep voltammetry test (Supplementary Fig. 8) and the floating voltage test (Supplementary Fig. 9), C2DP exhibits high anodic stability of up to 5.4 V vs. Li$^+$/Li, which can be ascribed to its irreversible pyridinium linkage and robust 2D polymer framework structure. The wide stable potential window of C2DP perfectly fits into the anion-intercalation process of graphite.

The C2DP-G electrode was then prepared by transferring C2DP onto the as-prepared graphite electrode as the electrode skin via an easy 'fishing' approach (Supplementary Fig. 10). Synthetic graphite with high purity (Supplementary Fig. 11) was utilized to prepare the electrode. No chemical bonding between C2DP and the graphite electrode was expected to form due to the inert polymer structure of C2DP. The initial interaction between C2DP and the graphite electrode is mainly van der Waals force and the possible electrostatic interaction. Both graphite and C2DP-G electrodes were assessed in two-electrode Swagelok cells with 2 M LiPF$_6$ in DMC as the electrolyte. Cyclic voltammetry curves of C2DP, the graphite electrode, and the C2DP-G electrode (Supplementary Fig. 12) confirm the negligible contribution of C2DP to the overall capacity of the C2DP-G electrode. Galvanostatic charge/discharge (GCD) curves at a voltage window of 3.5-5.1 V (Supplementary Fig. 13) were further collected to evaluate their rate performance (Fig. 2a). Compared with the graphite electrode (97 mAh g$^{-1}$ at 0.1 C, 60.8 mAh g$^{-1}$ at 20 C, 1 C = 100 mA g$^{-1}$), the C2DP-G electrode displays negligibly changed specific capacity (97 mAh g$^{-1}$ at 0.1 C) and rate capability (57.7 mAh g$^{-1}$ at 20 C), but significantly improved Coulombic efficiencies. This result suggests that the anion diffusivity degradation of the electrode, often induced by SEI[21,28], is not the case for the C2DP-G electrode.

The non-degraded capacities of the C2DP-G electrode can be explained by the superior anion transport of C2DP, which was assessed by loading C2DP onto a commercial polypropylene (PP) separator (denoted C2DP-PP) and performing electrochemical impedance spectroscopy (EIS) at variable temperatures (Supplementary Fig. 14). The ionic conductivity (σ, S cm$^{-1}$) was evaluated according to Eq. (1), where l (cm), R (Ω), and A (cm$^2$) correspond to the thickness, resistance, and area of PP or C2DP-PP, respectively. Impressively, C2DP-PP (e.g., 0.25 S m$^{-1}$ at 25 °C) depicts an ultrahigh ionic conductivity at the same order of magnitude as that of the PP separator (e.g., 0.63 S m$^{-1}$ at 25 °C). Of note, this ionic conductivity is substantially higher than Li$^+$-conductivities of the typical SEI in Li-ion batteries (~10$^{-7}$ S m$^{-1}$)[29] and classical solid-state electrolytes for Li-ion batteries (10$^{-6}$-10$^{-2}$ S m$^{-1}$)[31]. Fig 2b further plots log (σ) as a function of T$^{-1}$. The activation energy (E$_a$, eV) of ion transport is estimated from the curve slope in Fig. 2b based on Eq. (2), where T represents the absolute temperature (K), k is the Boltzmann constant (1.3806 × 10$^{-23}$ m$^2$ kg s$^{-2}$ K$^{-1}$), and b represents the pre-exponential factor. As calculated, C2DP-PP exhibits a low activation energy of 0.60 eV, suggesting the high ion-transport kinetics of C2DP. Moreover, we compared the anion (PF$_6^-$) transference number (t$_-$) of PP and C2DP-PP by the potentiostatic polarization test (Supplementary Fig. 15) and the EIS measurement (Supplementary Fig. 16) in symmetric Li//Li cells according to Eqs. (3) and (4). t$_+$ represents the Li$^+$ transference number, I$_O$ and I$_S$ are the initial and steady-state currents, ΔV is the potential applied across the cell (10 mV), R$_O$ and R$_S$ are the initial and steady-state interfacial resistances of the passivation layers on the Li electrode. Supplementary Table 1 summarizes the R$_O$ and R$_S$ values of both cells for PP and C2DP-PP. As expected, C2DP-PP presents a much higher t$_-$ (0.51) than PP alone (0.17), indicating the favorable anion-transport selectivity of C2DP

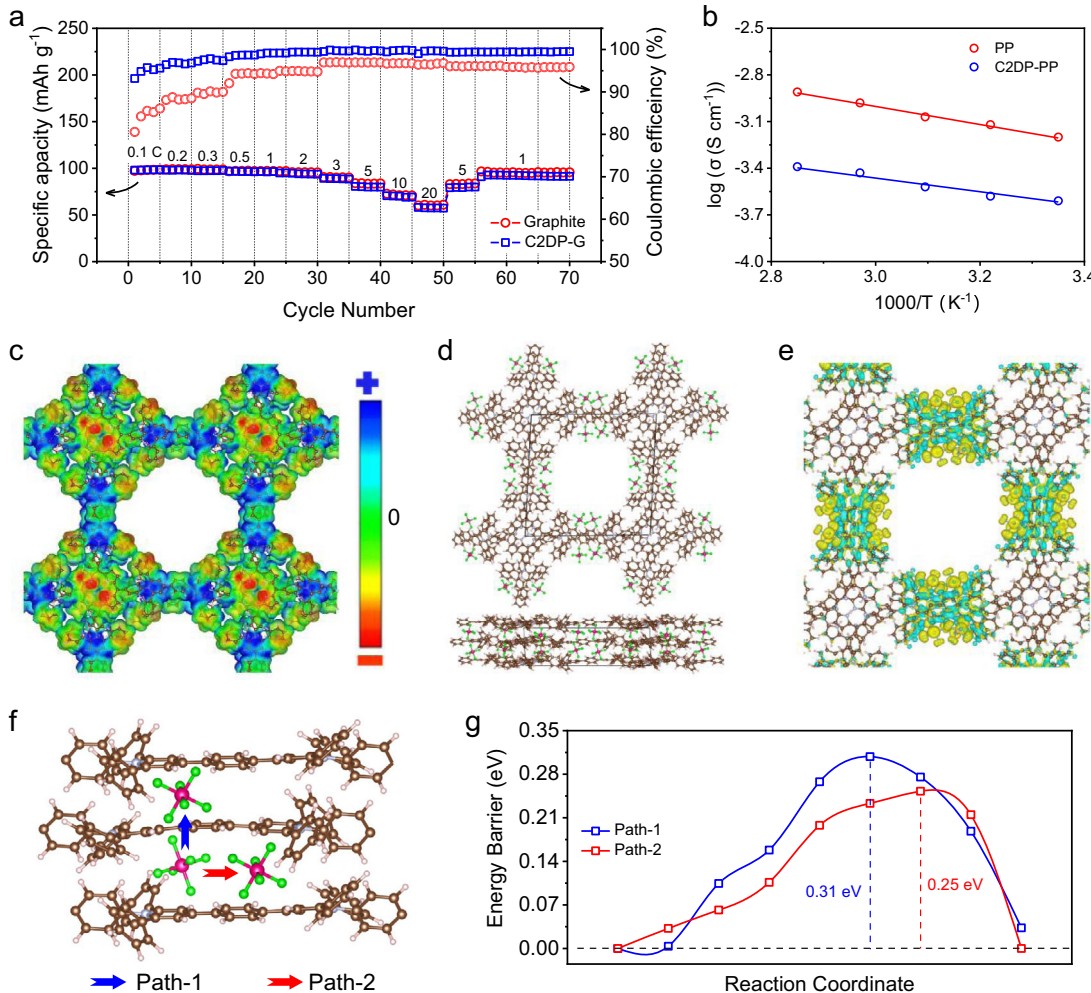

**Fig. 2 | Rate capability and PF$_6^-$ diffusivity. a** Rate performance of the graphite electrode and the C2DP-G electrode. **b** Log ($\sigma$) as a function of $T^{-1}$ for PP and C2DP-PP. **c** EPS plots of C2DP (isosurface value = 0.005 e Å$^{-3}$). The blue and red colours denote deficient and rich electron density, respectively. **d** Simulated structure and **e** charge density difference plots ($\rho_{adsorbed} - \sum\rho_{isolated}$, isosurface value = 0.0002 e Å$^{-3}$) of C2DP with PF$_6^-$ as counter ions. **f** Schematic illustration of PF$_6^-$-diffusion paths and **g** the corresponding PF$_6^-$-diffusion energy barriers.

$$\sigma = \frac{l}{R \times A} \quad (1)$$

$$\sigma = b \times exp\left(-\frac{E_a}{kT}\right) \quad (2)$$

$$t_+ = \frac{I_s(\triangle V - I_0 R_0)}{I_0(\triangle V - I_s R_s)} \quad (3)$$

$$t_- = 1 - t_+ \quad (4)$$

Density functional theory (DFT) calculations are carried out to simulate the PF$_6^-$-transport behavior of C2DP for in-depth insights. Evidenced by the electrostatic potential surface (EPS) plots of C2DP (Fig. 2c), the exposed pyridinium N atoms show relatively positive electrostatic potential (blue color), suggesting their function as anion adsorption and transportation sites. The most stable PF$_6^-$-compensated structure of C2DP among the various possible configurations (Supplementary Fig. 17) indicates that PF$_6^-$ prefers to be located in the pore of C2DP and adjacent to the two pyridinium N atoms. Fig 2d

illustrates the optimal C2DP unit cell with two serrated C2DP layers and eight PF$_6^-$ anions. Each PF$_6^-$ occupies the most stable adsorption site, and the minimum F-F distance between two adjacent PF$_6^-$ anions is in the range of 3.2-4.4 Å. The charge density difference plots (Fig. 2e) elucidate a net gain of electronic charge in the F atoms of PF$_6^-$ anions and a net loss of electronic charge in the pyridinium N sites of C2DP. Besides, the overall charge state of PF$_6^-$ (−0.95) is slightly higher than the formal charge state (−1), with the average atomic charges of +3.73 and −0.78 for P and F, respectively. This result indicates the inter-molecular electrostatic interaction induced by the cationic scaffold and charge-compensated PF$_6^-$ anions. Apart from the electrostatic interaction, C2DP layers are stacked by the weak interlayer van der Waals force, which is mainly induced by the aromatic porphyrin rings (Supplementary Fig. 18). Phenyl rings around tri-branched cationic centers are not in the same plane. The cationic pyridinium centers force C2DP layers to energetically stack in an inclined AA-stacking mode, thus avoiding cationic pyridinium centers directly on top of each other.

Two possible PF$_6^-$-diffusion paths, interlayer diffusion (*c*-direction, Path-1) and intralayer diffusion (*b*-direction, Path-2), are identified for C2DP (Fig. 2f). Supplementary Fig. 19 illustrates the detailed diffusion paths with transition states. The calculated energy barrier plots (Fig. 2g) indicate low diffusion barriers for both Path-1 (0.31 eV) and

Path-2 (0.25 eV), corresponding to the diffusion coefficients of $1.2 \times 10^{-7}$ cm² s⁻¹and $7.5 \times 10^{-7}$ cm² s⁻¹, respectively. The relationship between ionic conductivity and diffusivity is given by the Nernst-Einstein relationship (Eq. (5))[32], where F ($9.65 \times 10^4$ C mol⁻¹) is the Faraday constant, $c$ (mol L⁻¹) is the bulk molar salt concentration, R (8.31 J K⁻¹ mol⁻¹) is the gas constant, and D (m² s⁻¹) is the ion diffusion coefficient. Based on the measured ionic conductivity of C2DP-PP, the ion diffusion coefficient of C2DP-PP is estimated to be $5.6 \times 10^{-7}$ cm² s⁻¹, which is well consistent with the simulated value of C2DP. It should be emphasized that both the experimentally measured and theoretically calculated diffusion coefficients are several orders of magnitude higher than the experimentally determined PF₆⁻ diffusion coefficient ($10^{-12}$-$10^{-11}$ cm² s⁻¹) of graphite[33]. This finding supports that the ultra-thin C2DP membrane does not significantly limit the charge-storage kinetics of the graphite cathode.

$$D = \frac{\sigma RT}{F^2 c} \qquad (5)$$

## Alleviation of electrolyte decomposition

The calculated density of states reveal that C2DP possesses a direct band gap of 2.53 eV and all flat bands with no exception for valence band maximum and conduction band minimum (Supplementary Fig. 20). These flat bands suggest the very low electronic conductivity of C2DP, which is important to prevent the electrolyte decomposition on the C2DP-G electrode. To highlight the alleviation effect of C2DP on electrolyte decomposition, the first 40 charge/discharge cycles at 1 C were collected for both the graphite electrode and the C2DP-G electrode. As shown in Fig. 3a, the C2DP-G electrode exhibits substantially enhanced Coulombic efficiencies compared with the graphite electrode. In detail, the graphite electrode only achieves an initial Coulombic efficiency of 81.4%. Even after 40 charge/discharge cycles, the Coulombic efficiency only reaches 95.7%, which can hardly meet the standard for reliable practical applications. In striking contrast, the effective role of C2DP in avoiding the direct contact of graphite with the electrolyte enables the C2DP-G electrode with an impressive initial Coulombic efficiency of 92.0%. Moreover, the C2DP-G electrode exhibits superb Coulombic efficiencies of 96.2%, 98.8%, and 99.4% at the 5ᵗʰ, 10ᵗʰ, and 40ᵗʰ cycles, respectively (Fig. 3b).

The serious electrolyte decomposition causes the impedance rise of the graphite cathode, which can be identified by the electrochemical impedance spectroscopy measurement (Fig. 3c). The collected Nyquist plots were analyzed with the equivalent circuit (Supplementary Fig. 21), and the fitting result is shown in Supplementary Table 2. After 20 charge/discharge cycles, both interphase resistance ($R_{CEI}$) and

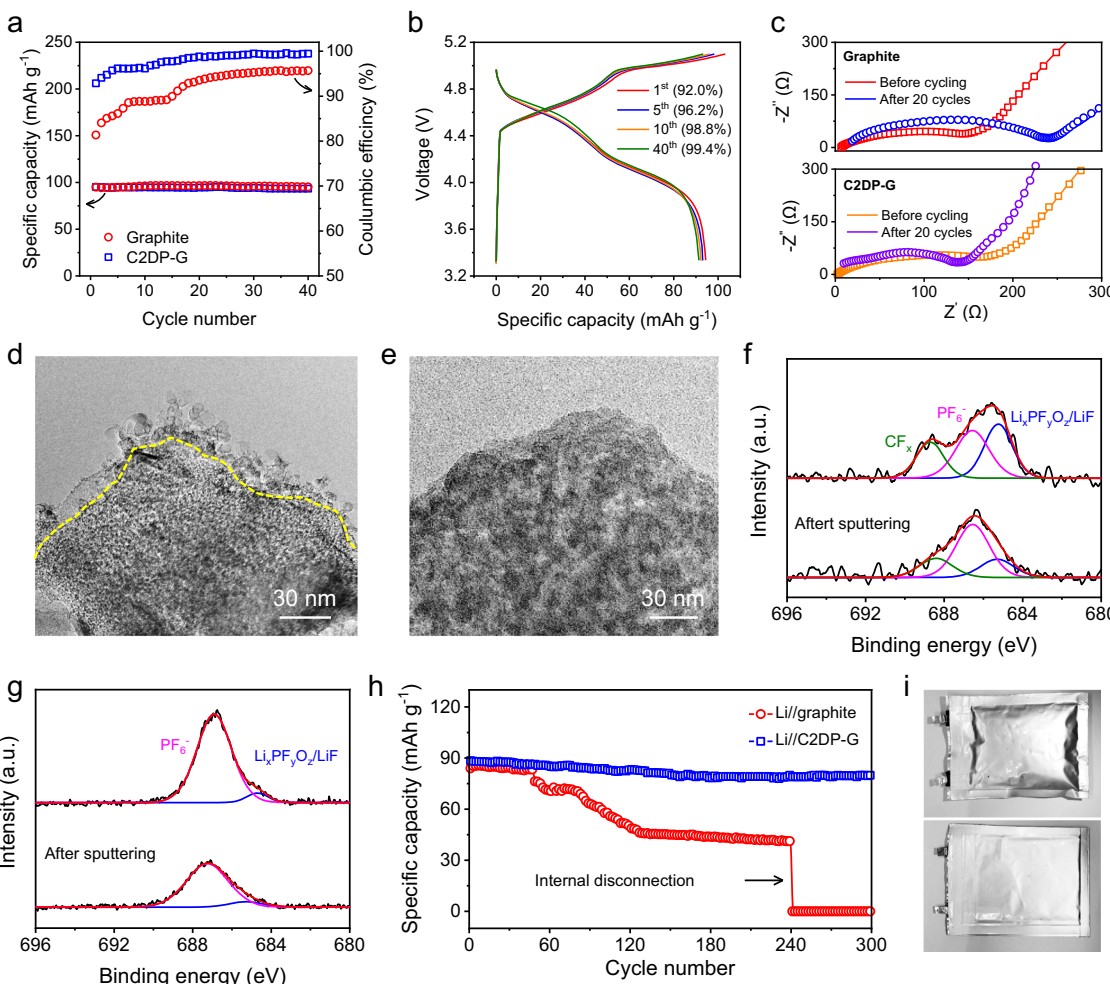

**Fig. 3 | Analysis of electrolyte decomposition. a** Specific capacities and Coulombic efficiencies of the graphite electrode and C2DP-G electrode in the initial 40 charge/discharge cycles at 1 C. **b** The 1ˢᵗ, 5ᵗʰ, 10ᵗʰ, 40ᵗʰ GCD curves of the C2DP-G electrode at 1 C. **c** Nyquist plots of the graphite electrode and the C2DP-G electrode before and after 20 GCD cycles. TEM images of the graphite grain in **d** the graphite electrode and **e** the C2DP-G electrode after 3 GCD cycles. **f, g** F 1 s XPS spectra of (**f**) the fully charged graphite electrode and (**g**) the fully charged C2DP-G electrode. **h**, Cycling performance of the Li//graphite pouch cell and the Li//C2DP-G pouch cell. **i** Digital photos of the Li//graphite pouch cell (upper) and the Li//C2DP-G pouch cell (lower) after the cycling test.

charge-transfer resistance ($R_{ct}$) of the graphite electrode experienced an apparent increase. In detail, $R_{CEI}$ increased from 7.4 Ω to 93.5 Ω, while $R_{ct}$ increased from 98.5 Ω to 142.6 Ω. This drastically increased resistance is dominantly assigned to the formed thick (10-20 nm) and amorphous interface on the graphite grain, which is clearly visible in the TEM image (Fig. 3d) and spectrally detectable by F K-edge X-ray absorption near edge structure spectra under the X-ray microscope (Supplementary Fig. 22). Notably, the thick interface on the graphite grain again evidences the undesirable electrolyte decomposition on the graphite electrode. In contrast, the $R_{CEI}$ of the C2DP-G electrode increased slightly from 10.7 Ω to 17.3 Ω after 20 charge/discharge cycles. The C2DP-G electrode before cycling showed a higher $R_{ct}$ (148.6 Ω) than the graphite electrode, which can be explained by the to-be-activated $PF_6^-$-wettability of C2DP. Significantly, the $R_{ct}$ of the C2DP-G electrode after the 20-cycle activation decreased to 101.3 Ω, close to the original $R_{ct}$ of the graphite electrode. Moreover, the graphite grain in the C2DP-G electrode after 20 charge/discharge cycles shows a clean surface, almost without a new interface (Fig. 3e).

Ar$^+$ sputtering-assisted X-ray photoelectron spectroscopy (XPS) was employed to probe into the composition of the formed interface on the graphite grains. To ensure that the XPS spectra of the C2DP-G electrode were measured for the graphite part instead of the C2DP layer, we carefully scratched the surface of the C2DP-G electrode with a doctor blade. Moreover, Ar$^+$ sputtering was conducted for 30 min, which could enable a sputtering depth of >45 nm. This sputtering process could further ensure the detection of the graphite part, as the C2DP thickness is around 20 nm. The fully charged graphite electrode surface depicts an intensive F 1s signal consisting of intercalated $PF_6^-$

(686.8 eV), $Li_xPF_yO_z$ (685.3 eV), and $CF_x$ (688.8 eV), with the latter two species predominant (Fig. 3f). Even after sputtering for 30 min, $Li_xPF_yO_z$ and $CF_x$ account for a considerable amount in the graphite electrode, which contrasts with the rare observation of these two species in the C2DP-G electrode (Fig. 3g). Besides, pronounced organic species were also detected by the O 1s and C 1s XPS spectra of the graphite electrode (Supplementary Fig. 23). All these inorganic/organic species formed on the graphite grain are known as poor anion transporter[8,14], thus potentially deteriorating the anion intercalation performance of the graphite electrode.

Another concern associated with electrolyte decomposition is the safety issue of battery devices owing to gas generation. The C2DP skin can effectively inhibit the gas formation on the graphite cathode, which was proved by assembling single-layer Li//graphite and Li//C2DP-G pouch cells. As shown in Fig. 3h, the Li//C2DP-G cell could stably operate for 300 charge/discharge cycles, but fast capacity decay was revealed for the Li//graphite cell after only 50 charge/discharge cycles. After the cycling test, gas release induced by the electrolyte decomposition swells the Li//graphite cells dramatically, while the gas generation of the Li//C2DP-G cell was clearly suppressed (Fig. 3i).

## Boosted durability of the anion-intercalation chemistry
We next assessed the durability of the graphite electrode and C2DP-G electrode by conducting the long-term cycling test at 1 C (Fig. 4a). It was found that the graphite electrode delivered a stable capacity for only around 300 cycles. The capacity started fading from the 300th cycle and decreased to 55% and 32% of the initial capacity at the 450th cycle and 1000th cycle, respectively. By contrast, the C2DP-G electrode

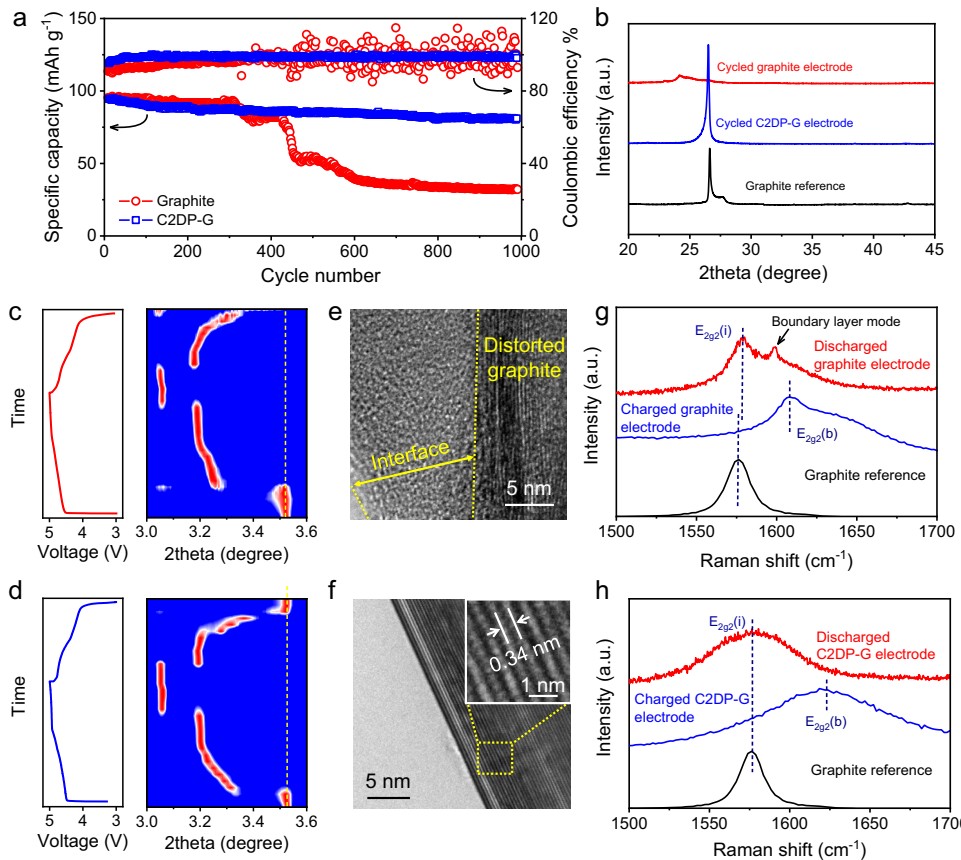

**Fig. 4 | Durability assessment of the $PF_6^-$-intercalation chemistry. a** Cycling performance of the graphite electrode and C2DP-G electrode at 1 C. **b** XRD patterns of the graphite electrode and the C2DP-G electrode after 1000 GCD cycles at 1 C. **c, d** Operando synchrotron XRD maps of (**c**) the graphite electrode and (**d**) the C2DP-G electrode during the initial cycle at 0.1 C. **e, f** TEM images of the graphite grain in (**e**) the graphite electrode and (**f**) the C2DP-G electrode after 20 GCD cycles. **g, h** Raman spectra of (**g**) the graphite electrode and (**h**) the C2DP-G electrode at both the fully charged and discharged states.

exhibited an impressive cycle life with high-capacity retention of 93% after 1000 cycles. It is notable that the durability of C2DP-G electrode represents the best among reported anion-intercalation chemistries of graphite in organic electrolytes (Supplementary Table 3). Fig 4b compares the XRD patterns of the graphite electrode and C2DP-G electrode after the cycling test (1000 cycles at 1 C). Compared with the graphite reference (26.7°), the C2DP-G electrode after cycling showed a (002) peak at 25.9°, corresponding to the slight interlayer distance expansion from 0.336 to 0.346 nm. By contrast, the lattice structure of the graphite electrode after cycling was completely deteriorated.

During the repeating charge/discharge process of the C2DP-G electrode, the C2DP layer was discovered to well maintain its mechanical and chemical structure (Supplementary Fig. 24 and Supplementary Fig. 25). To understand the origin of the graphitic structure collapse, operando synchrotron XRD (wavelength: 0.207 Å) during the initial charge/discharge cycle was carried out. The (002) peak of the graphite electrode after one charge/discharge cycle was located at 3.35° (Fig. 4c), unable to revert to the initial position (3.52°). This observation was early assigned to the co-intercalation of solvent[34,35], which accelerates the graphite lattice distortion. In contrast, the (002) peak of the C2DP-G electrode returned to 3.52° after anion de-intercalation (Fig. 4d), which is indicative of the excellent charge/discharge reversibility. Besides, the distorted structure and thick interface formed by electrolyte decomposition were clearly observed for the graphite electrode after 20 GCD cycles by the high-resolution TEM image (Fig. 4e), whereas the C2DP-G electrode after 20 GCD cycles (Fig. 4f) presented a smooth surface and a highly crystalline structure with the negligibly changed interlayer distance (0.340 nm).

Additionally, Fig. 4g and Fig. 4h compare the G-band Raman spectra ($E_{2g}$ vibration associated with the $sp^2$ carbon stretching) of the graphite electrode and C2DP-G electrode at both fully charged and discharged stages. At the charged stage, both electrodes depict the positively shifted G band ($E_{2g2}$(b), 1623 cm$^{-1}$), corroborating the anion intercalation into graphite. Unlike the C2DP-G electrode, the graphite electrode shows an additional shoulder peak at the fully discharged stage, which is known as the boundary layer mode[36]. The boundary layer mode indicates the partial stage II-like intercalation structure (each graphite layer is sandwiched by another graphite layer and an intercalated layer) in the graphite electrode, implying the confinement of solvent between graphite layers. This result manifests that C2DP

could effectively prevent the solvent co-intercalation into graphite and thus promote the durability of the C2DP-G electrode. From the simulated structure, the pore size of C2DP is determined about 2 nm (Fig. 2d), which agrees well with the high-resolution TEM image (Fig. 1e). In the LiPF$_6$ electrolyte, large solvated PF$_6^-$ (2-3 nm), solvated PF$_6^-$-Li$^+$ pairing (2-3 nm), and solvent-shared PF$_6^-$-Li$^+$ (3-4 nm) species were revealed by the early studies[37–39]. In this sense, the nano-sized pores exhibit the sieving function with the steric effect towards the big aggregated species. However, we should mention that the pore size of C2DP is not the only decisive factor. The cationic pyridinium centers are of great importance for the PF$_6^-$-selective transport, as they serve as the PF$_6^-$-hopping sites and assist in the dissociation of PF$_6^-$ with Li$^+$ and solvent.

## Universal effect for diverse anion-intercalation chemistries

Apart from the PF$_6^-$ intercalation chemistry, we found that C2DP as the electrode skin was also effective in promoting the reversibility and durability of various anion-intercalation chemistries (i.e., FSI$^-$ and TFSI$^-$). Both the graphite electrode and C2DP-G electrode were evaluated in 2 M LiFSI and 2 M LiTFSI electrolytes with the GCD test at 1 C. In 2 M LiFSI, the Coulombic efficiency of the C2DP-G electrode reaches 94.2%, 98.0%, 98.7%, and 99.2% at the 1st, 5th, 10th, and 40th cycles, respectively (Fig. 5a). These Coulombic efficiency values substantially surpass those of the graphite electrode (74.9%, 86.3%, 92.1%, and 97.4% at the 1st, 5th, 10th, and 40th cycle, respectively, Supplementary Figure 26a, b). Likely, the C2DP-G electrode achieves a high Coulombic efficiency of 99.0% at the 40th cycle in 2 M LiTFSI (Fig. 5b), while the graphite electrode only depicts a Coulombic efficiency of 97.1% (Supplementary Fig. 26c, d). Moreover, the boosted durability of the C2DP-G electrode is supported by the high capacity retention of 89.1% in 2 M LiFSI and 88.7% in 2 M LiTFSI after 1000 cycles, which contrasts with the pristine graphite electrode with low capacity retention of 56% in 2 M LiFSI (Fig. 5c) and 48% in 2 M LiTFSI (Fig. 5d). We also found that the C2DP layer could promote the ClO$_4^-$-intercalation chemistry of graphite, although the chemistry is still not feasible to demonstrate practical battery devices (Supplementary Fig. 27 and Supplementary Fig. 28). All these results manifest that C2DP possesses a universal protective effect for anion-intercalation chemistries, showing broad application opportunities in diverse anion-related battery devices.

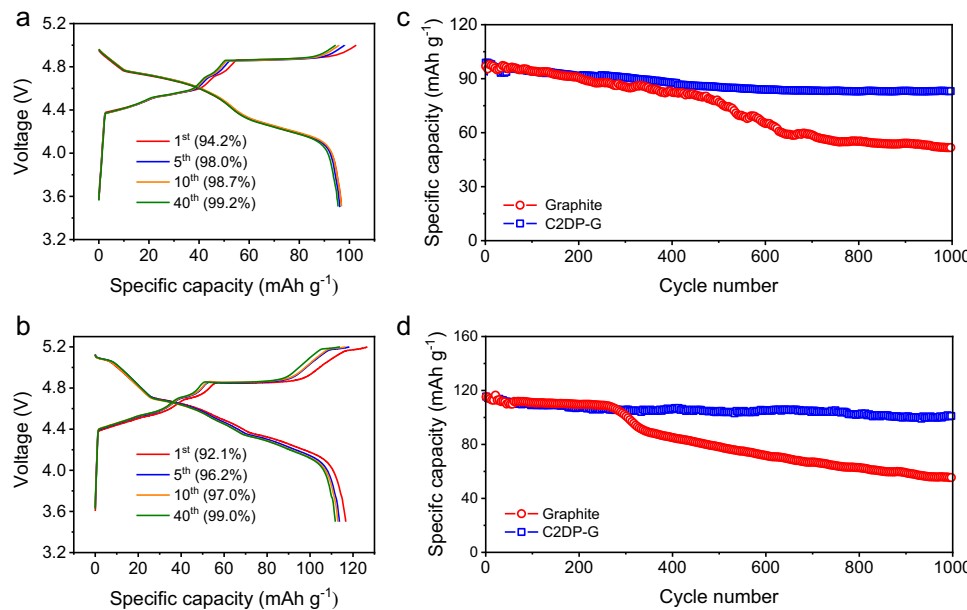

**Fig. 5 | Evaluation of other anion-intercalation chemistries. a, b** The 1st, 5th, 10th, 40th GCD cycles of the C2DP-G electrode in (**a**) 2 M LiFSI and (**b**) 2 M LiTFSI. **c, d** Cycling performance at 1 C of the graphite electrode and the C2DP-G electrode in (**c**) 2 M LiFSI and (**d**) 2 M LiTFSI.

## Discussion

In summary, we have demonstrated the ultrathin, cationic charged 2D polymer membrane as the effective artificial electrode skin to promote the anion-intercalation graphite chemistries. The ion-regulation capability of C2DP brought benefits from three aspects: (1) the positively charged backbone of C2DP enables fast anion transport through the electrode-electrolyte interface ($PF_6^-$ diffusivity: ~$10^{-7}$ cm$^2$ s$^{-1}$), thus empowering the C2DP-G electrode with non-declined capacity and anion-intercalation kinetics; (2) insulating C2DP physically isolates the electrolyte (particularly the solvent molecules) to reach the electrode, which effectively alleviates the electrolyte decomposition and improves the Coulombic efficiency of the anion-intercalation graphite chemistries; (3) the high anion-transport selectivity of C2DP ($PF_6^-$ transference number of PP *vs.* C2DP-PP, 0.17 *vs.* 0.51) inhibits the co-intercalation of solvent into the graphite lattice, protecting the graphite electrode from fast capacity degradation associated with the graphite structure collapse. The constructed C2DP-G electrode exhibited impressively enhanced durability for diverse anion-intercalation chemistries (e.g., $PF_6^-$, FSI$^-$, and TFSI$^-$), operating stably for 1000 cycles with the capacity retention of >85% and Coulombic efficiency of >99%. The demonstrated artificial electrode skin holds the possibility to be extended to other emerging organic/inorganic compounds relying on anion chemistries, further advancing the frontier of battery devices.

## Methods

### Chemicals

Synthetic graphite (<45 μm, 99.99%), LiPF$_6$ (battery grade, ≥99.99%), DMC (anhydrous, ≥99%), and alginic acid sodium salt (Alg, medium viscosity) were purchased from Sigma Aldrich. Super P, Li foil (~250 μm in thickness), and stainless steel mesh were bought from Alfa Aesar. Li(CF$_3$SO$_2$NSO$_2$CF$_3$)$_2$ (LiTFSI, 99.5%) and Li(FSO$_2$NSO$_2$F)$_2$ (LiFSI, 99.5%) were purchased from Solvionic. 5,10,15,20-(tetra-4-aminophenyl) porphyrin, 1,4-phenylene-4,4′-bis (2,6-diphenyl-4-pyrylium tetrafluoroborate), sodium oleyl sulfate (SOS) and solvents (e.g., chloroform) were purchased from PorphyChem and Sigma-Aldrich, and used directly without further purification. Water was purified using a Milli-Q purification system (Merck KGaA). All the reactions were carried out under an ambient atmosphere. Substrates (e.g., 300 nm SiO2/Si wafer, quartz glass and copper grids) were obtained from Microchemicals and Plano GmbH.

### Synthesis of C2DP

C2DP was synthesized through the surfactant monolayer-assisted interfacial synthesis method recently developed by us[27]. In detail, Milli-Q water (40 mL) was added into a beaker (80 mL, diameter = 6 cm). Afterwards, sodium oleyl sulfate (SOS) dissolved in chloroform (10 μL, 1 mg mL$^{-1}$) was spread onto the water surface. After 30 min, a mixed aqueous solution (1 mL) of compound 1 (7.9 × 10$^{-4}$ mmol) and CH$_3$COOH (1.75 mmol) was injected into the water phase, leading to a pH of 3.2. Compound 1 was attached underneath the SOS monolayer driven by the electrostatic interaction between the anionic head groups of SOS and the protonated compound 1. After 2 h, an aqueous solution of compound 2 (7.9 × 10$^{-4}$ mmol, 1 mL) was injected into the mixture. Triethylamine (TEA, 1.75 mmol) was next used to adjust the solution pH to 10.6. After another 2 h, the solution pH was tuned to 4.3 by adding CH$_3$COOH (8.75 mmol). The mixture was kept at 1 °C for 6 days for the polymerization reaction. Afterwards, the obtained purple C2DP membrane was transferred onto the substrates (e.g., SiO$_2$/Si wafer, TEM grid, and the graphite electrode) by the simple fishing method. The C2DP-loaded substrate was immersed in Milli-Q water for 5 min and rinsed with flowing ethanol, Milli-Q water. Finally, the sample was dried in N$_2$ flow.

## Characterizations

MALDI-TOF analysis was conducted with a Bruker Reflex II-TOF spectrometer using a 337 nm nitrogen laser. 2-((2E)-3-(4-*tert*-butylphenyl)-2-methylprop-2-enylidene)malononitrile was employed as the matrix. Optical microscopy (Zeiss), atomic force microscopy (AFM) (NT-MDT), and scanning electron microscopy (SEM, Zeiss Gemini 500) were used to investigate the morphological structure. C2DP was deposited on a 300 nm SiO2/Si substrate for the SEM, optical microscopy, and AFM characterization. FTIR spectra were measured by Tensor II (Bruker) with an attenuated total reflection (ATR) unit. TEM was conducted using a JEOL JEM F200 TEM instrument (200 kV). X-ray photoelectron spectroscopy (XPS) was performed using a multiprobe system (Scienta Omicron) with a monochromatized X-ray source (Al K$_\alpha$) and an electron analyzer (Argus CU). The spectra were fitted using Voigt functions after background subtraction and calibration using the C 1 s peak (C−C, 284.6 eV). For depth profiling, the samples were sputtered in the same multiprobe system using Ar$^+$ ions (FOCUS FDG150, 1 keV, 10 mA) for 30 min. To study the surface evolutions, the battery cells were disassembled in the glovebox, and the electrodes were washed with DMC for several times to remove the electrolyte residual. XRD measurements were taken at 2.2° per minute between 10 and 50.2° on an Analytical X'Pert Pro diffractometer with Cu-Kα radiation. Raman spectra were collected using a confocal Raman Microscope (S&I Monovista CRS + ). The samples were excited with a 640 nm laser (LASOS diode pumped solid state laser). The laser was aligned and then focused on the sample using an Olympus 20x objective with a laser power of 1 mW.

The GIWAXS measurements were carried out at the ID13 beamline at ESRF, Grenoble, France. It was conducted with a Dectris EigerX 4 M detector and the photon beam energy of 12.398 keV (λ = 1 Å). The beam incidence angle was ≈0.1°, and the sample-detector distance was verified to be 243 mm by using a chromium oxide calibration standard. X-ray absorption near edge structure (XANES) data was collected at beamline U41-TXM at the BESSY II electron storage ring. The collected spectra were subsequently processed with the ATHENA program to remove the background and apply appropriate normalization procedures[40].

Operando X-ray diffraction experiments were performed at beamline P02.1 at the PETRA III synchrotron (DESY, Hamburg, Germany). The experiments were carried out using coin cells with Kapton windows. The electrochemical measurement was conducted with a multichannel potentiostat VMP3 (Biologic). Each diffraction pattern was recorded every 10 min, while the cells were charged and discharged at 0.1 C. The electrolyte used was 2 M LiPF$_6$ in DMC. The graphite electrode and the C2DP-G electrode were used as the cathodes, and Li foil was used as the anode.

### Ionic behavior of C2DP

The ionic conductivity of C2DP at various temperatures was measured by electrochemical impedance spectroscopy (EIS) measurement. C2DP was transferred on the PP separator and then dried at room temperature for 48 h. C2DP-PP or PP was sandwiched between two stainless steel plates in a Swagelok cell. EIS measurement was measured with a potential amplitude of 10 mV and frequencies from 100 kHz to 1 Hz at various temperatures on the electrochemical instrument (multichannel potentiostat VMP3, Biologic). Li$^+$ transference number ($t_+$) of C2DP was derived by the chronoamperometry method. C2DP-PP or PP was sandwiched between two Li foil in a Swagelok cell. $t_+$ was calculated from the Vincent-Evans equation (Eq. (3)), where $I_0$ and $I_s$ are the initial and steady-state currents, $\Delta V$ is the potential applied across the cell (10 mV), $R_O$ and $R_S$ are the initial and steady-state interfacial resistances of the passivation layers on the Li electrode. $PF_6^-$ transference number ($t_-$) of C2DP was further determined by Eq. (4)

## Battery assembly

The graphite electrode was prepared by a traditional slurry-coating method. Typically, a slurry mixture consisting of 70 wt% graphite, 20 wt% carbon black, and 10 wt% alginate acid sodium salt was cast onto an Al foil and then dried at 120 °C overnight under vacuum. The mass loading of the active materials on the electrode was around 2 mg cm$^{-2}$. The C2DP-G electrode was obtained by simply transferring C2DP on the graphite electrode via the fishing method. 2 M LiPF$_6$, LiFSI, or LiTFSI dissolved in DMC was used as the electrolyte. Swagelok cells were assembled in an Ar-filled glove box using the Celgard 2400 separator and the Li metal anode. All the electrodes and separators for Swagelok cells have a diameter of 10 mm. Pouch cells were assembled by using the graphite electrode or the C2DP-G electrode as a cathode (the cathode size: 55 mm × 40 mm), Li foil as an anode (the anode size: 40 mm × 20 mm). Graphite was coated on the Al foil with an areal density of 15-20 mg cm$^{-2}$. Subsequently, the electrodes were compacted and then dried at 65 °C under an Ar atmosphere for 72 h. Li foil was pressed on Stainless steel mesh and used as the anode.

## Electrochemical measurements

All the electrochemical measurements were carried out in a lab with the constant temperature (25 °C). Cyclic voltammetry, linear sweep voltammetry, and floating voltage test with the chronoamperometry method were measured on a CHI electrochemical workstation (CHI 760E). The battery performance was measured on the LAND battery testing system. We defined 100 mA g$^{-1}$ as 1 C, because the discharge time of our electrodes at 100 mA g$^{-1}$ was close to 1 h (0.98 h for the graphite electrode, 0.96 h for the C2DP-G electrode). The voltage windows used for the measurements in the LiPF$_6$, LiTFSI, and LiFSI electrolytes are 3-5.1 V, 3-5.2 V, and 3-5.0 V, respectively.

## Theoretical calculations

The first-principles calculations as implemented in the Vienna Ab initio Simulation Package (VASP version 5.4.4)[41,42] were used for all the calculations. The exchange-correlation potential was described by using a generalized gradient approximation of Perdew-Burke-Ernzerhoff. The projector augmented-wave method was used to impose the interaction between ion cores and valence electrons[43–45]. A 470 eV cut-off of plane-wave basis was used for structure optimization. The DFT-D3 extension of Grimme was adopted to add the vdW correction for potential energy and interatomic forces48. The energy and force convergence criteria were set as 10$^{-5}$ eV and 0.01 eV Å$^{-1}$ for structural optimization and energy calculation. For C2DP, all the calculations were carried out with the serrated AA stacked bulk structure. In the calculations of charged C2DP, the total electron number of the whole system was governed, and then the jellium background charge was introduced automatically to maintain charge neutrality. A further dipole correction was added to localize the positive charge. In the case of the fully PF$_6^-$-compensated C2DP unit cell (C2DP with 8 PF$_6^-$), the charge neutrality was balanced by the counter PF$_6^-$ anions, which have the charge state of −0.95, close to the formal charge state (−1). The Brillouin zone was sampled with a Gamma centred k-point grid of 1 × 1 × 3 for C2DP and 3 × 3 × 3 for graphite. The climbing image nudged elastic band (CI-NEB) method was used to locate the diffusion pathway[46]. The minimum energy paths (MEP) were initialized by considering seven image structures between initial and final structural geometries, and the energy convergence criterion of each image was set as 10$^{-5}$ eV. Activation barriers were calculated by the energy differences between the transition and initial states. Spin polarization was included in all calculations. Visualization of the structures was made using VESTA software. The diffusion constant is roughly estimated as Eq. (6), where ν is the effective vibrational frequency (typically, 10$^{13}$ s$^{-1}$)[47], $a$ is the hopping distance, $E_a$ is the activation energy, $k_B$ is the Boltzmann

constant, and T = 300 K is the temperature.

$$D = \nu^* a^2 e^{\left(\frac{-E_a}{k_B T}\right)} \qquad (6)$$

## Data availability

The authors declare that all the relevant data are available within the paper and its Supplementary Information file or from the corresponding authors upon reasonable request.

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

## Acknowledgements

This work was financially supported by European Union's Horizon 2020 research and innovation programme (GrapheneCore3 881603, LIGHT-CAP 101017821), M-ERA.NET and Sächsisches Staatsministerium für Wissenschaft und Kunst (HYSUCAP 100478697 & Sonderzuweisung zur Unterstützung profilbestimmender Struktureinheiten), and German Research Foundation (DFG) within the Cluster of Excellence, CRC 1415 (Grant No. 417590517), and Polymer-based Batteries (SPP 2248, RACOF-MMIS). P.B. acknowledges the Alexander-von-Humboldt foundation for funding. R.D. appreciates the funding support from ERC starting grant (FC2DMOF, No. 852909). C.N. and A.T. acknowledge financial support of the DFG through a research infrastructure grant INST 275/357-1 FUGG and the Thüringer Ministerium für Wirtschaft, Wissenschaft und Digitale Gesellschaft (TMWWDG) and the Thüringer Aufbau Bank (TAB) within the project LiNaKon (2018 FGR 0092). The authors acknowledge the use of the facilities in the Dresden Center for Nanoanalysis (DCN) at the Technische Universität Dresden, the GWK support for providing computing time through the Center for Information Services and High-Performance Computing (ZIH) at TU Dresden, beam time allocation at beamline P02.1 at the PETRA III synchrotron (DESY, Hamburg, Germany), beam time allocation at beamline ID13 of the European Synchrotron Radiation Facility (ESRF), and the Helmholtz-Zentrum Berlin für Materialien und Energie for the allocation of synchrotron radiation beamtime (beamline U41-TXM). The authors thank Dr. Rishi Shivhare and Prof. Stefan Mannsfeld for GIWAXS measurement.

## Author contributions

X.F., M.Y., and D.S. conceived the overall project. D.S. and Z.W. designed the procedure under the supervision of M.Y. and R.D., and they performed the dominant experiments and data analysis. P.H. assisted in material characterization and interpretation by SEM. P.B. and T.H. performed the DFT calculations. Q.L. and D.M. performed in-situ XRD experiment. C.N. and A.T. performed the XPS characterization. A.S.N. conducted the XPS analysis. Z.L., D.L., and E.Z. collected TEM images. A.M.D. performed the Raman characterization. A.M. conducted the preparation and evaluation of the pouch cell. All authors read and commented on the manuscript.

## Funding

## Competing interests

The authors declare no competing interests.
