## [Peer Review File · Nature Communications]

Ultrathin positively charged electrode skin for durable anion-intercalation battery chemistriesREVIEWER COMMENTS

Reviewer #1 (Remarks to the Author):

This work describes the use of ultrathin positively charged two-dimensional poly(pyridinium salt) membrane (denoted C2DP) as the artificial skin for the graphite cathode, to tackle the inherent reversibility and durability issues of the anion-intercalation chemistries. Large-area C2DP (28 cm²), fabricated by an upscalable on-water surface synthesis approach, that can be conformally coated on the graphite electrode. the C2DP-covered graphite cathode (denoted C2DP-G) achieves an impressive PF₆⁻-intercalation durability with the capacity retention of 92.8% after 1,000 cycles at 1 C and Coulombic efficiencies of > 99%, which contrasts with the pronounced performance degradation of the pristine graphite cathode after 300 cycles. Overall the research work presented in this article is important and can be considered for publication after some revisions.

1. In figure S9, What is the possible explanation for changing discharge voltage plateaus at high current densities? Can the authors provide some explanation on the shape-changing for the discharge curve?
2. How 1 C rate was defined? Was it defined on the base of discharge time or simply literature?
3. A CV of only C2DP against Lithium could be added to see any interference or contribution to the capacity.
4. Figure S10, An equivalent circuit could be added., that is used for EIS calculations. Also, in the EIS discussion, SEI layer resistance should also be included.
5. How robust was C2DP after cycling, a digital image would be nice to add.
6. Figure 4b, After how many cycles XRD was performed? Is it ok to say the negligibly expanded interlayer distance? As XRD shifts to the lower theta.
7. Please, correct the caption to supplementary figure 17 a-d, as b & d do not present data for the C2DP-G electrode.
8. A brief description of what type of graphite? As graphite microstructure, crystallinity, and defect density influence anion intercalation and its reversibility. Please, mention /provide some details (Raman/XRD) of the pristine graphite powder.
9. In the Ionic behavior of C2DP, how the values of R₀ and R_S these values of resistances were quantified?

Reviewer #2 (Remarks to the Author):

This work reports an ultrathin protection layer for graphite cathode, which enables better anion intercalation behaviour. Different anion intercalation behaviours, such as PF₆⁻, FSI⁻, and TFSI⁻ into the graphite in assist of the C2DP layer are studied. Overall, the much-promoted CE and cycling stability of graphite cathodes are impressive. And the working mechanism of the protective layer is well demonstrated. However, the data interpretation is oversimplified, and the novelty of this work is not well presented. Therefore, major revisions should be taken before the acceptance.

1. I'd suggest the authors re-organize the Introduction part since the decomposition of the electrolyte is routed in the instability of the electrolyte molecular upon high voltage and/or the exposed active graphite edge in the electrolyte. Also, the advantages of the as-prepared C2DP layer should be highlighted in this section.
2. The authors evaluated different anion intercalation chemistries of the C2DP-G electrode, such as PF₆⁻, FSI⁻, and TFSI⁻. All these three anions contain F elements. Are there any specific criteria for selecting a suitable anion for this C2DP layer? Is this C2DP layer suitable to promote the intercalation behaviour of ClO₄⁻?
3. The characterization of the C2DP layer after cycling should be provided. Since the C2DP functions as a protection layer, the chemical/physical properties before and after cycling are crucial for evaluating its practical feasibility. Also, is the protective layer present when characterizing the cycled C2DP-G electrodes?? This should be clarified in the manuscript for better understanding.
4. From the XPS of the cycled C2DP-graphite electrode, electrolyte decomposition products can be identified. It is confusing whether these species formed on the surface of graphite electrodes or the C2DP protection layer. Also, the pronounced difference in the C 1s spectra of the fully charged graphite electrode and fully charged C2DP-G electrode is noticed. C=O signal vanished in the spectra of the C2DP-G electrode. Any explanations?

5. Any TEM images showing the thickness of the C2DP layer on the graphite electrode?
6. Look at the simulated stacking configuration of the C2DP layer, What is the intermolecular force between the C2DP molecules?
7. The determination of PF6⁻ transference number needs detailed discussion. "Given the strong Li⁺ selectivity of PP, the PF6⁻ transference number of C2DP is thus estimated to be close to 0.85" such a statement is not scientific.
8. Is there any interaction between the C2DP layer and the graphite electrode?
9. The caption of Figure s17 is unclear.

Reviewer #3 (Remarks to the Author):

In this work, the authors designed a positively charged two-dimensional poly (pyridinium salt) membrane (denoted C2DP) as the graphite electrode skin to overcome the critical durability problem of DIBs. The large-area C2DP membrane enables the conformal coating on the graphite electrode, alleviating the electrolyte decomposition and the formation of the anion-blocking passivation layer. The C2DP empowers the C2DP-G electrode with non-declined capacity and fast anion-intercalation kinetics. As a result, the C2DP-G electrode can obtain improved capacity retention of 92% after 1000 cycles at 1 C and high CE of over 99%. The study offers viable means to promote the problematic DIBs chemistries that rely on complex charge carriers. However, some critical information in the current work is missing. Many key issues are unclear and need to be further clarified. Therefore, I think some significant improvements are required before further review. My detailed comments are as follows:

1. The authors provided a positively charged 2D C2DP as the graphite electrode skin to enhance the reversibility of graphite. However, in Fig.3(a), only 40 cycles are selected to demonstrate the advantages of C2DP for improving the CE. The CE with more cycle numbers should be provided to confirm this issue.
2. The authors claimed the C2DP exhibits high anodic stability up to 5.6 V using LSV. However, the oxidation voltage is highly related to the scan rate. Scan rate should be provided. Additionally, recently, a floating test has been used as a more reliable indicator for the voltage stability test, so it is recommended to perform a floating test in which the voltage stability can be more accurately compared.
3. How about the yield rate of C2DP? Is there any side reaction during the preparation of C2DP? If there are side reactions, how to separate and purify them?
4. What is the size of the C2DP pore? Pore size is also vital for ion sieving. The authors should carefully investigate this issue and give a detailed discussion.
5. The authors claimed only the PF6⁻ XPS peak can be observed in the C2DPG electrode after sputtering. What is the sputtering depth? The XPS results are heavily related to depth. To better confirm this issue, the authors should conduct an XPS test after peeling the C2DPG skin.
6. To illustrate the robustness and stability of the C2DPG, the microstructure of the C2DPG skin after cycling should be investigated using SEM and AFM.
7. From XRD results in Fig. 4b, it can be revealed that the lattice structure of the graphite electrode was completely deteriorated, as also mentioned by the authors. However, obvious lattice fringes can still be observed in Fig.4e. The authors should give reasonable explanations.
8. In terms of methodology, evaluating intermolecular interaction energies using DFT methods is notoriously difficult, and those energies are essential to this study, especially considering the anion group of PF6⁻. How do the authors calculate the energy of a charged system in their calculation?
9. The authors claimed that the C2DP is electron-insulating. Some theoretical evidence should be provided.
10. The configurations of C2DP with one PF6⁻ in Supplementary Figure 12 are analogous. It is difficult to distinguish them. The authors should provide several explicit configurations to replace them.

Reviewer #4 (Remarks to the Author):

In this manuscript, Sabaghi et al. carry out a combined theoretical and experimental investigation of a charged two-dimensional poly(pyridinium salt) membrane (in short, C2DP) as protective

graphite electrode skin. In this report, I will only assess the theoretical part of the manuscript since that is my area of expertise. The authors have conducted density functional theory (DFT) based simulations of the diffusion of PF₆⁻ anion in the C2DP membranes. They used state-of-the-art methods, i.e., they employed dispersion corrections in their calculations. They analyze the preferred adsorption sites of PF₆⁻ in C2DP and how this anion hops from site to site using the nudged elastic band (NEB) method. They found a theoretical diffusion coefficient very close to the experimental one, which validates their model. Thus, at least from a theoretical point of view, I can recommend this paper for publication in Nature Communications.

To Reviewer 1:

This work describes the use of ultrathin positively charged two-dimensional poly (pyridinium salt) membrane (denoted C2DP) as the artificial skin for the graphite cathode, to tackle the inherent reversibility and durability issues of the anion-intercalation chemistries. Large-area C2DP (28 cm²), fabricated by an up scalable on-water surface synthesis approach, that can be conformally coated on the graphite electrode. the C2DP-covered graphite cathode (denoted C2DP-G) achieves an impressive PF6⁻-intercalation durability with the capacity retention of 92.8% after 1,000 cycles at 1 C and Coulombic efficiencies of > 99%, which contrasts with the pronounced performance degradation of the pristine graphite cathode after 300 cycles. Overall, the research work presented in this article is important and can be considered for publication after some revisions.

Response: We appreciate the positive comment of the reviewer. Additional experiments and revisions have been conducted according to your following comments.

1. In Figure S9, what is the possible explanation for changing discharge voltage plateaus at high current densities? Can the authors provide some explanation on the shape-changing for the discharge curve?

Figure R1. GCD curves of **a** the graphite electrode and **b** the C2DP-G electrode at different rates.

Response: Thank you for your insightful question. **Figure R1** shows galvanostatic charge-discharge (GCD) profiles of both graphite and C2DP-G electrodes at various rates. At low rates (0.1–3 C), the curves nearly overlap with each other, and only slight voltage polarization (voltage gap between the charge and discharge plateaus) and capacity decay were detected. At high rates (5 C and 10 C), the discharge plateaus, particularly the high-voltage plateau (>4.4 V), become sloped with apparently decreased voltage. This shape change comes from the large voltage polarization (the combination of

activation polarization, ohmic polarization, and concentration polarization) induced by the high rate. The corresponding discussion has been added into the revised manuscript.

2. How 1 C rate was defined? Was it defined on the base of discharge time or simply literature?

Response: We defined 100 mA g^{-1} as 1 C, because the theoretical specific capacity of graphite with PF_6^- intercalation was reported to be between 93 mAh g^{-1} (C_{24}PF_6) and 112 mAh g^{-1} (C_{20}PF_6)¹. Moreover, in our case, the discharge time of our electrodes at 100 mA g^{-1} was close to 1 h (0.98 h for the graphite electrode, 0.96 h for the C2DP-G electrode). We have clarified it in our revised manuscript.

3. A CV of only C2DP against lithium could be added to see any interference or contribution to the capacity.

Response: According to your suggestion, a CV curve of C2DP loaded on the Al foil was collected in 2 M LiPF_6 . **Figure R2** compares the CV curves of C2DP, the graphite electrode, and the C2DP-G electrode at 1 mV s^{-1} . Apparently, C2DP shows negligible contribution to the overall capacity of the C2DP-G electrode. The corresponding discussion has been added into the revised manuscript.

Figure R2. CV curves of C2DP, the graphite electrode, and the C2DP-G electrode at 1 mV s^{-1} .

4. Figure S10, an equivalent circuit could be added, that is used for EIS calculations. Also, in the EIS discussion, SEI layer resistance should also be included.

Response: Thank you for your constructive advice. The equivalent circuit (**Figure R3**) has been added for Supplementary Figure 10. Moreover, we have revised the EIS discussion by including the resistance relevant to the SEI layer. The detailed discussion can be found in the revised manuscript or below.

Figure R3. Equivalent circuit for the EIS fitting.

Figure R4. Nyquist plots of the graphite electrode **a** before and **b** after 20 GCD cycles. Nyquist plots of the C2DP-G electrode **c** before and **d** after 20 GCD cycles.

“The serious electrolyte decomposition causes the impedance rise of the graphite cathode, which can be identified by the electrochemical impedance spectroscopy measurement (**Figure R4**). The collected Nyquist plots were analyzed with the equivalent circuit (**Figure R5**), and the fitting result is shown in **Table R1**. After 20 charge/discharge cycles, both interphase resistance (R_{CEI}) and charge-transfer resistance (R_{ct}) of the graphite electrode experienced an apparent increase. In detail, R_{CEI} increased from 7.4Ω to 93.5Ω , while R_{ct} increased from 98.5Ω to 142.6Ω .”

Figure R5. Equivalent circuit for the EIS fitting.

“In contrast, the R_{CEI} of the C2DP-G electrode increased slightly from 10.7 Ω to 17.3 Ω after 20 charge/discharge cycles. The C2DP-G electrode before cycling showed a higher R_{ct} (148.6 Ω) than the graphite electrode, which can be explained by the to-be-activated PF_6^- -wettability of C2DP. Significantly, the R_{ct} of the C2DP-G electrode after the 20-cycle activation decreased to 101.3 Ω , close to the original R_{ct} of the graphite electrode.”

Table R1. The EIS fitting result of the graphite electrode and the C2DP-G electrode before and after 20 GCD cycles.

Electrode	Electrolyte resistance (R_e)	Interphase resistance (R_{CEI})	Charge-transfer resistance (R_{ct})
Graphite before cycling	2.7 Ω	7.4 Ω	98.5 Ω
Graphite after 20 cycles	5.8 Ω	93.5 Ω	142.6 Ω
C2DP-G before cycling	1.8 Ω	10.7 Ω	148.6 Ω
C2DP-G after 20 cycles	3.4 Ω	17.3 Ω	101.3 Ω

5. How robust was C2DP after cycling, a digital image would be nice to add.

Response: According to your suggestion, the SEM image of the C2DP-G electrode after 100 charge/discharge cycles was collected. **Figure R6** compares the SEM images of the graphite electrode and the C2DP-G electrode after 100 GCD cycles. The continuous C2DP layer was clearly identified on the surface of the C2DP-G electrode, supporting the strong robustness of C2DP in the battery environment. The corresponding discussion has been added into the revised manuscript.

Figure R6. SEM images of **a** the graphite electrode and **b** the C2DP-G electrode after 100 GCD cycles.

6. Figure 4b, after how many cycles XRD was performed? Is it ok to say the negligibly expanded interlayer distance? As XRD shifts to the lower theta.

Response: Fig. 4b shows the XRD patterns of the graphite electrode and the C2DP-G electrode after 1,000 charge/discharge cycles at 1 C. Compared with the graphite reference (26.7°), the C2DP-G electrode after cycling showed a (002) peak at 25.9° , corresponding to the slight interlayer distance expansion from 0.336 to 0.346 nm. By contrast, the lattice structure of the graphite electrode after cycling was completely deteriorated. We have changed our description in the revised manuscript accordingly.

7. Please, correct the caption to Supplementary Figure 17 a-d, as b & d do not present data for the C2DP-G electrode.

Response: Thank you for your kind reminder. The caption of Supplementary Figure 17 has been corrected in the revised manuscript. The revised caption is

“**a** Coulombic efficiencies of the graphite electrode and the C2DP-G electrode in 2 M LiFSI. **b** GCD curves of the graphite electrode in 2 M LiFSI. **c** Coulombic efficiencies of the graphite electrode and the C2DP-G electrode in 2 M LiTFSI. **d** GCD curves of the graphite electrode in 2 M LiTFSI.”

8. A brief description of what type of graphite? As graphite microstructure, crystallinity, and defect density influence anion intercalation and its reversibility. Please, mention /provide some details (Raman/XRD) of the pristine graphite powder.

Figure R7. a XRD and b Raman spectra of graphite.

Response: In this study, synthetic graphite (Sigma Aldrich, <45µm, 99.99% trace metals basis) was utilized. According to your suggestion, XRD and Raman spectra of graphite were collected (**Figure R7**). The sharp characteristic XRD peaks at 26.7° for the (002) plane and 57.3° for the (004) plane indicate the high crystallinity of graphite with an interlayer spacing of 0.336 nm (**Figure R7a**). In the Raman spectrum (**Figure R7b**), graphite shows three prominent characteristic peaks, namely the D band, the G band, and the 2D band. The sharp G band indicates the highly ordered graphite layers arranged in AB Bernal stacking. These characteristic Raman peaks again evidence the high crystallinity with rare carbon defects. The corresponding discussion has been added into the revised manuscript.

9. In the ionic behavior of C2DP, how the values of R_0 and R_s these values of resistances were quantified?

Response: The Li^+ transference number (t_+) of PP and C2DP-PP was estimated according to **equation R1**. The initial resistance (R_0) and steady-state resistance (R_s) were derived by the EIS measurement of the symmetric cells (*i.e.*, Li//PP//Li and Li//C2DP-PP//Li) before and after polarization, respectively (**Figure R8**). **Table R2** summarizes the R_0 and R_s values of both cells. The detailed discussion has been added into the revised manuscript.

$$t_+ = \frac{I_s(\Delta V - I_0 R_0)}{I_0(\Delta V - I_s R_s)} \quad (\text{R1})$$

Figure R8. Nyquist plots of **a** the Li//PP//Li cell and **b** the Li//C2DP-PP//Li cell.

Table R2. The R_0 and R_s values of Li//PP//Li and Li//C2DP-PP//Li.

Type of cell	R_0	R_s
Li//PP//Li	498 Ω	401 Ω
Li//C2DP-PP//Li	1230 Ω	1158 Ω

To Reviewer 2

This work reports an ultrathin protection layer for graphite cathode, which enables better anion intercalation behaviour. Different anion intercalation behaviours, such as PF_6^- , FSI^- , and TFSI^- into the graphite in assist of the C2DP layer are studied. Overall, the much-promoted CE and cycling stability of graphite cathodes are impressive. And the working mechanism of the protective layer is well demonstrated. However, the data interpretation is oversimplified, and the novelty of this work is not well presented. Therefore, major revisions should be taken before the acceptance.

Response: We appreciate the valuable comments of the reviewer. Additional experiments and discussions have been conducted to address the following concerns.

1. I'd suggest the authors re-organize the Introduction part since the decomposition of the electrolyte is routed in the instability of the electrolyte molecular upon high voltage and/or the exposed active graphite edge in the electrolyte. Also, the advantages of the as-prepared C2DP layer should be highlighted in this section.

Response: Thank you for your constructive suggestion. The introduction part has been changed in the revised manuscript accordingly. The details can also be found below.

“There are two main reasons for the unsatisfactory battery performance (**Fig. 1a**). The electrolyte molecules tend to be oxidized at the exposed active graphite edge under the high voltage, causing the serious electrolyte decomposition and the formation of thick and resistant cathode-electrolyte interphase²⁻⁴. Meanwhile, cation/solvent co-intercalation, together with the gas release due to the electrolyte decomposition, leads to the fast structural degradation of the graphite cathode¹⁸⁻²⁰.”

“Specifically, large-area C2DP (28 cm²), fabricated by a facile and upscalable on-water surface synthesis approach, can be conformally coated on the graphite electrode. Like SEI, electron-insulating C2DP provides a passivation layer at the electrode-electrolyte interface, avoiding electrolyte decomposition and the consequent formation of the anion-blocking layer. Meanwhile, the densely equipped single-crystal domains with an ultrathin thickness (20 nm), a well-defined cationic backbone, and quasi-1D nanochannels confer C2DP with a high anion-transport capability (PF_6^- diffusivity: $\sim 10^{-7}$ cm² s⁻¹) and selectivity (PF_6^- transference number increased from 0.17 for the polypropylene separator to 0.51 for the C2DP-loaded polypropylene separator). When used as the graphite electrode skin, C2DP effectively screens cation/solvent from intercalating into graphite and thus suppresses the structure collapse of graphite grains. Besides, the ‘soft’ polymerized structure with irreversible pyridinium

linkages empowers C2DP with good chemical stability and mechanical robustness to adapt to the complex battery environment and withstand the electrode volume change.”

2. The authors evaluated different anion intercalation chemistries of the C2DP-G electrode, such as PF_6^- , FSI^- , and TFSI^- . All these three anions contain F elements. Are there any specific criteria for selecting a suitable anion for this C2DP layer? Is this C2DP layer suitable to promote the intercalation behavior of ClO_4^- ?

Response: In our study, PF_6^- , FSI^- , and TFSI^- were selected because their reversible intercalation chemistries into graphite were verified by the early effort^{2,5-8}. The attractive stage-I intercalation can be achieved by all three anions, reaching the attractive high redox potential (~ 4.5 V vs. Li^+/Li on average) and large specific capacity (>100 mAh g^{-1}). Containing F atoms is not a selective criterion. We believe the positively charged C2DP should exhibit the universal effect for diverse anions, including F-free ones.

Figure R9. The 1st, 2nd, 5th, and 10th GCD profiles of **a** the graphite electrode and the C2DP-G electrode in 2 M LiClO_4 dissolved in propylene carbonate at 0.3 C and 50 °C.

In fact, the ClO_4^- -intercalation chemistry of graphite was demonstrated to be not reversible and efficient by the early study.⁹ At an elevated temperature of 50 °C, ClO_4^- only achieved the stage-V intercalation with a low specific capacity (below 15 mAh g^{-1}) and Coulombic efficiency (below 25%). To address your concern, we have further studied the graphite electrode and the C2DP-G electrode in an electrolyte of 2 M LiClO_4 dissolved in propylene carbonate at 50 °C. **Figure R9** compares the GCD profiles of both electrodes at 0.3 C. Like the early study, the graphite electrode in our case depicted a low specific capacity of 7 mAh g^{-1} and low Coulombic efficiencies of 10.2% at the initial cycle and 24% at the 10th cycle. By contrast, the C2DP-G electrode reached a higher specific capacity of 21 mAh g^{-1} and enhanced Coulombic efficiencies of 59.5% at the initial cycle and 65.0% at the 10th cycle. The XRD patterns were

also collected for the discharged electrodes after 10 GCD cycles (**Figure R10**). Several side peaks beside the (002) plane were detected in the graphite electrode, which can be assigned to confinement of the aggregated intercalation compounds (*i.e.*, complexes of Li^+ , ClO_4^- , and propylene carbonate) between the graphite layers. However, these side peaks were not observed in the C2DP-G electrode. All these results indicate that the C2DP layer also works to promote the ClO_4^- -intercalation chemistry of graphite, although the chemistry is still not feasible to demonstrate practical battery devices. We have added the corresponding discussion into the revised manuscript.

Figure R10. XRD patterns of the fully discharged graphite electrode and the fully discharged C2DP-G electrode after 10 GCD cycles at 0.3 C in 2 M LiClO_4 dissolved in propylene carbonate and 50 °C.

3. The characterization of the C2DP layer after cycling should be provided. Since the C2DP functions as a protection layer, the chemical/physical properties before and after cycling are crucial for evaluating its practical feasibility. Also, is the protective layer present when characterizing the cycled C2DP-G electrodes? This should be clarified in the manuscript for better understanding.

Response: According to your suggestion, we further collected the SEM images, FTIR, and Raman spectra of the C2DP-G electrode before and after 100 charge/discharge cycles at 2 C. **Figure R11** compares the SEM images of the graphite electrode and the C2DP-G electrode after 100 GCD cycles. The continuous C2DP layer was clearly identified on the surface of the C2DP-G electrode, supporting the strong robustness of C2DP in the battery environment. Besides, **Figure R12** compares the FTIR and Raman spectra of C2DP and the C2DP-G electrode before and after 100 GCD cycles. Apparently, the C2DP-G electrode shows almost the same peaks before and after cycling. All the characteristic peaks of C2DP were well reserved by the cycled C2DP-G electrode. All these results verify the robust structural

stability of C2DP and the desirable feasibility for practical battery application. The corresponding discussion has been added into the revised manuscript.

Figure R11. SEM images of **a** the graphite electrode and **b** the C2DP-G electrode after 100 GCD cycles.

Figure R12. **a** FTIR and **b** Raman spectra of C2DP and the C2DP-G electrode before and after 100 GCD cycles at 2 C.

4. From the XPS of the cycled C2DP-G electrode, electrolyte decomposition products can be identified. It is confusing whether these species formed on the surface of graphite electrodes or the C2DP protection layer. Also, the pronounced difference in the C 1s spectra of the fully charged graphite electrode and fully charged C2DP-G electrode is noticed. C=O signal vanished in the spectra of the C2DP-G electrode. Any explanations?

Response: Thank you for the insightful question. We should stress that only rare electrolyte decomposition products were formed in the C2DP-G electrode, which can be supported by the almost

negligible F 1s XPS signal associated with the electrolyte decomposition product (**Fig. 2g**) and the clean graphite surface even after 1,000 charge/discharge cycles as observed from the TEM image (**Fig. 4f**). In the O 1s and C 1s XPS spectra of the fully charged C2DP-G electrode, signals associated with O species could come from the used electrode binder (*i.e.*, alginate acid sodium salt) and the carbon black additive. To ensure that the XPS spectra of the C2DP-G electrode were measured for the graphite part instead of the C2DP layer, we carefully scratched the surface of the C2DP-G electrode with a doctor blade. Moreover, we did Ar⁺ sputtering for 30 min, which generally enables a sputtering depth of > 45 nm. This sputtering process could further ensure the detection of the graphite part, as the C2DP thickness is around 20 nm.

Figure R13. **a** O 1s and **b** C 1s XPS spectra of the fully charged graphite electrode. **c** O 1s and **d** C 1s XPS spectra of the fully charged C2DP-G electrode.

In the C 1s spectra (**Figure R13**), deconvoluted peaks at 291.5 eV (C-O₃,C-F₃), 288.7 eV (COOH), and 287.9 eV (C=O) can be ascribed to the electrolyte decomposition products.¹⁰ Meanwhile, the peak at 285.3 eV (C-O/amorphous carbon) comes from the combination of the electrolyte decomposition

products, the binder, and the carbon black additive. The vanished C=O signal in the C2DP-G electrode evidences the suppressed electrolyte decomposition on the C2DP-G electrode. A similar conclusion can also be drawn by comparing the O 1s spectra of the fully charged graphite electrode and the fully charged C2DP-G electrode¹¹. The corresponding discussion has been added into the revised manuscript.

5. Any TEM images showing the thickness of the C2DP layer on the graphite electrode?

Response: We attempted to detect the C2DP layer thickness through the cross-sectional TEM image. Unfortunately, we found the typical focused ion beam process damaged the C2DP structure heavily. We failed to directly measure the thickness of the C2DP layer on the electrode. However, the crystalline domain thickness of C2DP before transferring onto the graphite electrode was measured to be 20 nm by atomic force microscopy (Fig. 1c). We believe that the C2DP layer should keep the same thickness after transferring onto the graphite electrode.

6. Look at the simulated stacking configuration of the C2DP layer, what is the intermolecular force between the C2DP molecules?

Figure R14. Atomic configurations of C2DP **a** with and **b** without charge-compensated PF_6^- .

Response: Thank you for the valuable question. From the simulated structure of C2DP, two intermolecular forces can be identified, namely, interlayer van der Waals interaction and the electrostatic interaction induced by the cationic scaffold and charge-compensated anions. Like other layered-stacked 2D materials, C2DP layers are stacked by the weak interlayer van der Waals force mainly induced by the aromatic porphyrin rings, while phenyl rings around tri-branched cationic centers are not in the same

plane (**Figure R14a**). Meanwhile, the cationic pyridinium centers force C2DP layers to energetically stack in an inclined AA-stacking mode, thus avoiding cationic pyridinium centers directly on top of each other. Moreover, the structure shows that the charge-compensated PF_6^- anions prefer to be located in the pore of C2DP and adjacent to the two pyridinium N atoms. Compared with non- PF_6^- structure (4.13 Å, **Figure R14b**), PF_6^- -containing C2DP has a slightly larger interlayer distance of 4.59 Å, further supporting the intermolecular electrostatic interaction. The corresponding discussion has been added into the revised manuscript.

7. The determination of PF_6^- transference number needs detailed discussion. “Given the strong Li^+ selectivity of PP, the PF_6^- transference number of C2DP is thus estimated to be close to 0.85” such a statement is not scientific.

Response: We agree with the reviewer and attempted to directly measure the Li^+ and PF_6^- transference number of C2DP by assembling the symmetric $\text{Li}/\text{C2DP}/\text{Li}$ cell. Unfortunately, it didn’t work because the ultrathin thickness of C2DP easily caused the short circuit of the cell. To avoid confusion, we have changed our description in the revised manuscript. Details can also be seen below.

“As expected, C2DP-PP presents a much higher PF_6^- transference number (0.51) than PP alone (0.17), indicating the favorable anion-transport selectivity of C2DP enabled by its positively charged framework.”

8. Is there any interaction between the C2DP layer and the graphite electrode?

Response: C2DP was transferred from the water surface onto the graphite electrode by a simple ‘fishing’ approach. In this sense, no chemical bonding between C2DP and the graphite electrode was expected to form due to the inert polymer structure of C2DP. The initial interaction between C2DP and the graphite electrode is mainly van der Waals force and the possible electrostatic interaction. These interactions can ensure the strong attachment of C2DP onto the graphite electrode, just like other 2D materials as the coating layers^{12,13}. During the repeating charge/discharge cycles, C2DP still reserved its robust chemical structure, as evidenced by **Figure R11** and **Figure R12**. Thereby, strong chemical bonding was also not expected to form between C2DP and the graphite electrode. Nevertheless, we could not exclude the possible slight covalent bonding between C2DP and graphite or between C2DP and the electrode binder, given the complicated battery environment (multiple components, including electrolyte and electrode species) and the imposed electric field. The interfacial bonding with the low quantity is challenging to be experimentally proved. We have added the corresponding discussion into the revised manuscript.

9. The caption of Figure S17 is unclear.

Response: Thank you for your kind reminder. The caption of Supplementary Figure 17 has been corrected in the revised manuscript. The revised caption can also be seen below.

“**a** Coulombic efficiencies of the graphite electrode and the C2DP-G electrode in 2 M LiFSI. **b** GCD curves of the graphite electrode in 2 M LiFSI. **c** Coulombic efficiencies of the graphite electrode and the C2DP-G electrode in 2 M LiTFSI. **d** GCD curves of the graphite electrode in 2 M LiTFSI.”

To Reviewer 3

In this work, the authors designed a positively charged two-dimensional poly (pyridinium salt) membrane (denoted C2DP) as the graphite electrode skin to overcome the critical durability problem of DIBs. The large-area C2DP membrane enables the conformal coating on the graphite electrode, alleviating the electrolyte decomposition and the formation of the anion-blocking passivation layer. The C2DP empowers the C2DP-G electrode with non-declined capacity and fast anion-intercalation kinetics. As a result, the C2DP-G electrode can obtain improved capacity retention of 92% after 1000 cycles at 1 C and high CE of over 99%. The study offers viable means to promote the problematic DIBs chemistries that rely on complex charge carriers. However, some critical information in the current work is missing. Many key issues are unclear and need to be further clarified. Therefore, I think some significant improvements are required before further review. My detailed comments are as follows:

Response: We appreciate the positive comment of the reviewer. Additional experiments and discussions have been conducted to address the following concerns.

1. The authors provided a positively charged 2D C2DP as the graphite electrode skin to enhance the reversibility of graphite. However, in Fig.3a, only 40 cycles are selected to demonstrate the advantages of C2DP for improving the CE. The CE with more cycle numbers should be provided to confirm this issue.

Figure R15. Cycling performance of the graphite electrode and the C2DP-G electrode at 1 C.

Response: According to your suggestion, the Coulombic efficiency data have been added into the revised Fig. 4a, which shows the cycling performance of the graphite electrode and the C2DP-G

electrode at 1 C for 1,000 cycles (**Figure R15**). The effect of C2DP on improving the reversibility of the C2DP-G electrode can be clearly identified.

2. The authors claimed the C2DP exhibits high anodic stability up to 5.6 V using LSV. However, the oxidation voltage is highly related to the scan rate. Scan rate should be provided. Additionally, recently, a floating test has been used as a more reliable indicator for the voltage stability test, so it is recommended to perform a floating test in which the voltage stability can be more accurately compared.

Response: Thank you for the constructive suggestion. We carried out the floating voltage test of the stainless-steel current collector (denoted SS) and C2DP-loaded SS (denoted C2DP-SS) with the chronoamperometry method in 2 M LiPF₆ dissolved in DMC (**Figure R16**). Four different voltages were employed, *i.e.*, 5.0 V, 5.2 V, 5.4 V, and 5.6 V. For SS, the leaking current density at 5.0 V was only 0.013 mA cm⁻², while it increased to 0.014 mA cm⁻², 0.018 mA cm⁻² and 0.026 mA cm⁻² at 5.2 V, 5.4 V, and 5.6 V, respectively. This result indicates the pronounced electrolyte decomposition reaction on SS at 5.4 V and 5.6 V. By contrast, C2DP-SS behaved almost the same at 5.0 V, 5.2 V, and 5.4 V, showing the fast current relaxation (< 11 s) and the low leaking current density of 0.0052 mA cm⁻². This result indicates that C2DP can completely inhibit electrolyte decomposition at 5.4 V owing to its electron-insulating property. When the voltage reached 5.6 V, although C2DP-SS showed a longer current relaxation time (300 s), the final leaking current is still around 0.005 mA cm⁻². This observation implies that C2DP can obviously suppress electrolyte decomposition even at 5.6 V. The corresponding discussion has been added into the revised manuscript.

Figure R16. The floating voltage test of **a** SS and **b** C2DP-SS with the chronoamperometry method at different voltages.

3. How about the yield rate of C2DP? Is there any side reaction during the preparation of C2DP? If there are side reactions, how to separate and purify them?

Response: As described in the Methods section, the synthesis of C2DP adopted a surfactant monolayer-assisted interfacial synthesis approach, which was extensively optimized to reach a high yield rate of >95 %. We did the model Katritzky reaction with 5-(4-aminophenyl)-10,15,20-(triphenyl)porphyrin and 2,4,6-triphenylpyrylium tetrafluoroborate under the similar reaction condition on the water surface. The performed matrix-assisted laser desorption/ionization–time-of-flight mass spectrometry (MALDI-TOF MS) result of the formed product on the water surface indicates the presence of the pyridinium target without detectable by-products (**Figure R17**). This result indicates the high reaction selectivity of Katritzky reaction in our interfacial synthesis approach without the presence of side reactions. After the polymerization, the synthesized C2DP film was deposited onto the substrates (*e.g.*, SiO₂/Si wafer, TEM grid and the graphite electrode) by the fishing method. To clean C2DP, the substrate with the C2DP film was immersed in Milli-Q water for 5 min and rinsed with flowing ethanol, Milli-Q water, and then dried in N₂ flow. From the FTIR (Supplementary Figure 5) and Raman (Supplementary Figure 6) spectra of C2DP, no impurities were evidenced. The related discussion has been added into the revised manuscript.

Figure R17. MALDI-TOF MS analysis of the model reaction on the water surface.

4. What is the size of the C2DP pore? Pore size is also vital for ion sieving. The authors should carefully investigate this issue and give a detailed discussion.

Response: Thank you for your insightful suggestion. The pore size of C2DP determined from the simulated structure is about 2 nm (**Figure R18a**), which was also experimentally supported by the high-resolution TEM (**Figure R18b**). In the LiPF₆ electrolyte, large solvated PF₆⁻ (2~3 nm), solvated PF₆⁻-Li⁺ pairing (2~3 nm), and solvent-shared PF₆⁻-Li⁺ (3~4 nm) species were revealed by the early studies¹⁴⁻¹⁶. In this sense, the nano-sized pores exhibit the sieving function with the steric effect towards the big

aggregated species. However, we should mention that the pore size of C2DP is not the only decisive factor. The cationic pyridinium centers are of great importance for the PF_6^- -selective transport, as they serve as the PF_6^- -hopping sites and assist in the dissociation of PF_6^- with Li^+ and solvent. The corresponding discussion has been added into the revised manuscript.

Figure R18. **a** Atomic structure of C2DP. **b** High-resolution TEM image of C2DP with the structural model overlaid and cation.

5. The authors claimed only the PF_6^- XPS peak can be observed in the C2DP-G electrode after sputtering. What is the sputtering depth? The XPS results are heavily related to depth. To better confirm this issue, the authors should conduct an XPS test after peeling the C2DP-G skin.

Response: Thank you for your constructive suggestion. To ensure that the XPS spectra of the C2DP-G electrode were measured for the graphite part instead of the C2DP layer, we carefully scratched the surface of the C2DP-G electrode with a doctor blade. Moreover, we did Ar^+ sputtering for 30 min, which generally enables a sputtering depth of > 45 nm. This sputtering process could further ensure the detection of the graphite part, as the C2DP thickness is around 20 nm. We have clarified this issue in the revised manuscript.

6. To illustrate the robustness and stability of the C2DP-G, the microstructure of the C2DP-G skin after cycling should be investigated using SEM and AFM.

Response: According to your suggestion, we further collected the SEM images, FTIR, and Raman spectra of the C2DP-G electrode before and after 100 charge/discharge cycles at 2 C. **Figure R19** compares the SEM images of the graphite electrode and the C2DP-G electrode after 100 GCD cycles.

The continuous C2DP layer was clearly identified on the surface of the C2DP-G electrode, supporting the strong robustness of C2DP in the battery environment. Besides, **Figure R20** compares the FTIR and Raman spectra of C2DP and the C2DP-G electrode before and after 100 GCD cycles. Apparently, the C2DP-G electrode shows almost the same peaks before and after cycling. All the characteristic peaks of C2DP were well reserved by the cycled C2DP-G electrode. All these results verify the robust structural stability of C2DP and the desirable feasibility for practical battery application. We should also mention that it is impossible to peel off C2DP from the cycled C2DP-G electrode, and thus it is challenging to conduct the AFM measurement of C2DP after cycling. The corresponding discussion has been added into the revised manuscript.

Figure R19. SEM image of the graphite and C2DP-G electrode after 100 charge/discharge cycles at 2 C.

Figure R20. **a** FTIR and **b** Raman spectra of C2DP and the C2DP-G electrode before and after 100 charge/discharge cycles at 2 C.

7. From XRD results in Fig. 4b, it can be revealed that the lattice structure of the graphite electrode was completely deteriorated, as also mentioned by the authors. However, obvious lattice fringes can still be observed in Fig. 4e. The authors should give reasonable explanations.

Response: We are sorry for the confusion. In fact, **Fig. 4e** and **Fig. 4f** compare the TEM images of the graphite grain in the graphite electrode and the C2DP-G electrode after 20 GCD cycles. **Fig. 4b** shows the XRD patterns of the graphite electrode and the C2DP-G electrode after 1,000 GCD cycles. We have clarified it in the revised manuscript.

8. In terms of methodology, evaluating intermolecular interaction energies using DFT methods is notoriously difficult, and those energies are essential to this study, especially considering the anion group of PF_6^- . How do the authors calculate the energy of a charged system in their calculation?

Response: The charged systems (*e.g.*, C2DP and C2DP with 1 PF_6^-) were studied by setting the total number of valence electrons and assuming a homogeneous background charge at the same. Along with this, a further dipole correction was added in order to localize the positive charge. To confirm the correct localization, we explored the electrostatic potential (ESP) plots (**Figure R21**), which show that the positive charges (denoted by blue colour) are precisely localized on pyridium nitrogen atoms. In the case of the fully PF_6^- -compensated C2DP unit cell (C2DP with 8 PF_6^-), the charge neutrality was balanced by the counter PF_6^- anions, which have the charge state of -0.95 , close to the formal charge state (-1). We have added the related discussion into the revised manuscript.

Figure R21. ESP plots of C2DP (isosurface value = $0.005 e \text{ \AA}^{-3}$).

9. The authors claimed that C2DP is electron-insulating. Some theoretical evidence should be provided.

Response: Thank you for your insightful suggestion. To evaluate the electron-transport property of C2DP, we calculated the HSE06 (Herd–Scuseria–Ernzerhof hybrid) functional density of states (DOSs, **Figure R22a**) and the Perdew–Burke–Ernzerhof band structure of C2DP (**Figure R22b**). Here, the computationally expensive HSE06 method provides a comparatively accurate assessment of the band gap, and the PBE band structure can precisely explain the nature of bands. It was revealed that C2DP possessed a direct band gap of 2.53 eV. On a closer look, the band structure consists of all flat bands with no exception for valence band maximum (VBM) and conduction band minimum (CBM). These flat VBM and CBM bands are associated with holes and electrons with infinite effective masses and zero velocity. Thus, the electrons/holes are "localized" or stuck around particular spatial locations, suggesting the very low hole/electron conductivity of C2DP. The corresponding discussion has been added into the revised manuscript.

Figure R22. **a** The HSE06 DOSs and **b** the PBE band structure of C2DP.

10. The configurations of C2DP with one PF_6^- in Supplementary Figure 12 are analogous. It is difficult to distinguish them. The authors should provide several explicit configurations to replace them.

Response: Thank you for pointing out the figure issue. We have replaced the figure with more explicit images in the revised manuscript. Details can also be found in **Figure R23**.

Figure R23. Top and side views showing various configurations of C2DP with one PF_6^- anion, including **a** site-1, **b** site-2, **c** site-3, **d** site-4, and **e** site-5. Site-4 was calculated as the most thermodynamically stable configuration.

To Reviewer 4

In this manuscript, Sabaghi *et al.* carry out a combined theoretical and experimental investigation of a charged two-dimensional poly (pyridinium salt) membrane (in short, C2DP) as protective graphite electrode skin. In this report, I will only assess the theoretical part of the manuscript since that is my area of expertise. The authors have conducted density functional theory (DFT) based simulations of the diffusion of PF₆⁻ anion in the C2DP membranes. They used state-of-the-art methods, i.e., they employed dispersion corrections in their calculations. They analyze the preferred adsorption sites of PF₆⁻ in C2DP and how this anion hops from site to site using the nudged elastic band (NEB) method. They found a theoretical diffusion coefficient very close to the experimental one, which validates their model. Thus, at least from a theoretical point of view, I can recommend this paper for publication in Nature Communications.

Response: We appreciate the positive comment of the reviewer.

References:

1. Read, J. A. In-situ studies on the electrochemical intercalation of hexafluorophosphate anion in graphite with selective cointercalation of solvent. *J. Phys. Chem. C* **119**, 8438–8446 (2015).
2. Qi, X. *et al.* Investigation of PF₆⁻ and TFSI⁻ anion intercalation into graphitized carbon blacks and its influence on high voltage lithium ion batteries. *Phys. Chem. Chem. Phys.* **16**, 25306–25313 (2014).
3. Yang, K. *et al.* Revealing the anion intercalation behavior and surface evolution of graphite in dual-ion batteries via in situ AFM. *Nano Res.* **13**, 412–418 (2020).
4. Zheng, Q. *et al.* A cyclic phosphate-based battery electrolyte for high voltage and safe operation. *Nat. Energy* **5**, 291–298 (2020).
5. Wu, S., Zhang, F. & Tang, Y. A Novel Calcium-Ion Battery Based on Dual-Carbon Configuration with High Working Voltage and Long Cycling Life. *Adv. Sci.* **5**, 1701082 (2018).
6. Li, X., Ou, X. & Tang, Y. 6.0 V High-Voltage and Concentrated Electrolyte toward High Energy Density K-Based Dual-Graphite Battery. *Adv. Energy Mater.* **10**, 2002567 (2020).
7. Beltrop, K. *et al.* Enabling bis (fluorosulfonyl) imide-based ionic liquid electrolytes for application in dual-ion batteries. *J. Power Sources.* **373**, 193–202 (2018).
8. Placke, T. *et al.* Reversible Intercalation of Bis(trifluoromethanesulfonyl)imide Anions from an Ionic Liquid Electrolyte into Graphite for High Performance Dual-Ion Cells. *J. Electrochem. Soc.* **159**, A1755–A1765 (2012).
9. Gao, J., Tian, S., Qi, L. & Wang, H. Intercalation manners of perchlorate anion into graphite electrode from organic solutions. *Electrochim. Acta* **176**, 22–27 (2015).
10. Li, W. H. *et al.* Highly Improved Cycling Stability of Anion De-/Intercalation in the Graphite Cathode for Dual-Ion Batteries. *Adv. Mater.* **31**, 1804766 (2019).
11. Han, X. *et al.* An In Situ Interface Reinforcement Strategy Achieving Long Cycle Performance

- of Dual-Ion Batteries. *Adv. Energy Mater.* **9**, 1804022 (2019).
12. Thi, Q. H., Kim, H., Zhao, J. & Ly, T. H. Coating two-dimensional MoS₂ with polymer creates a corrosive non-uniform interface. *npj 2D Mater. Appl.* **2**, 34 (2018).
 13. Owji, E., Mokhtari, H., Ostovari, F., Darazereshki, B. & Shakiba, N. 2D materials coated on etched optical fibers as humidity sensor. *Sci. Rep.* **11**, 1771 (2021).
 14. Pal, B., Yang, S., Ramesh, S., Thangadurai, V. & Jose, R. Electrolyte selection for supercapacitive devices: A critical review. *Nanoscale Adv.* **1**, 3807–3835 (2019).
 15. Winter, M. & Placke, T. Does Size really Matter? New Insights into the Intercalation Behavior of Anions into a Graphite- Based Positive Electrode for Dual-Ion Batteries. *Electrochim. Acta.* **13**, 4686 (2016).
 16. Huang, Z. *et al.* Manipulating anion intercalation enables a high-voltage aqueous dual ion battery. *Nat. Commun.* **12**, 3106 (2021).

REVIEWERS' COMMENTS

Reviewer #2 (Remarks to the Author):

My comments were well addressed. Overall, in this work, the positively charged thin membrane is utilized as the protective skin for the graphite electrode. The provided theoretical and experimental data demonstrate that the membrane skin enables a stable interface and exhibits high anion-transport velocity and selectivity. As a result, it is impressive that prolonged durability for 1000 cycles at 1 C has been achieved. This work undoubtedly offers an innovative template for durable anion-intercalation battery chemistries. Therefore, I recommend its publication in Nature Communications with no further comments.

Reviewer #3 (Remarks to the Author):

The issues have been well addressed by the authors. It could be accepted at present.

To Reviewer 2

My comments were well addressed. Overall, in this work, the positively charged thin membrane is utilized as the protective skin for the graphite electrode. The provided theoretical and experimental data demonstrate that the membrane skin enables a stable interface and exhibits high anion-transport velocity and selectivity. As a result, it is impressive that prolonged durability for 1000 cycles at 1 C has been achieved. This work undoubtedly offers an innovative template for durable anion-intercalation battery chemistries. Therefore, I recommend its publication in Nature Communications with no further comments.

Response: We sincerely thank the referee for the valuable evaluation.

To Reviewer 3

The issues have been well addressed by the authors. It could be accepted at present.

Response: We sincerely thank the referee for the valuable evaluation.